# On the Shelf Life of Fine-Tuned LLM-Judges: Future-Proofing, Backward-Compatibility, and Question Generalization

**Janvijay Singh**[1,2*] **Austin Xu**[1†] **Yilun Zhou**[1†] **Yefan Zhou**[1,3*]
**Dilek Hakkani-Tür**[2] **Shafiq Joty**[1]

[1]Salesforce AI Research    [2]University of Illinois Urbana-Champaign    [3]Dartmouth College

## Abstract

The LLM-as-a-judge paradigm is widely used in both evaluating free-text model responses and reward modeling for model alignment and fine-tuning. Recently, fine-tuning judges with judge-specific data has emerged as an often preferred choice over directly prompting frontier models as judges, as the former achieves better performance with smaller model sizes while being more robust to common biases. However, the standard evaluation ignores several practical concerns of fine-tuned judges regarding their real-world deployment. In this paper, we identify and formalize three aspects that affect the *shelf life* of these judges: *future-proofing* and *backward-compatibility* – how well judges fine-tuned on responses by today's generator models perform on responses by future models or past models, as well as *question generalization* – how well judges generalize to unseen questions at test time. We study these three aspects under a unified framework with varying train and test distributions in two reasoning datasets, three SFT- and DPO-based fine-tuning algorithms, and three different backbone models. Experiments suggest that future-proofing is challenging for most models, while backward-compatibility is relatively easy, with DPO-trained models consistently *improving* performance. We further find that continual learning provides a more balanced adaptation to shifts between older and newer response distributions than training solely on stronger or weaker responses. Moreover, all models exhibit some degree of performance degradation when moving from questions seen during training to unseen ones, showing that current judges do not fully generalize to unseen questions. These findings provide insights into practical considerations for developing and deploying judge models in the face of ever-changing generators.

**Code:** https://github.com/iamjanvijay/judge-training-analysis

## 1 Introduction

Automatic evaluators have become a central part of the large language model (LLM) development cycle. They serve both as reward models during training (Stiennon et al., 2020; Ouyang et al., 2022; Yuan et al., 2024) and as verifiers in inference-time compute scaling (Zhou et al., 2025; Kim et al., 2025; Singhi et al., 2025). In the LLM-as-judge paradigm, a generative language model evaluates the outputs of other models for a given input question, providing a scalable approach to automatic evaluation. Past work on LLM-as-judges began with zero-shot prompting of capable LLMs (Liu et al., 2023; Dubois et al., 2023). However, such judges have been shown to be prone to various biases, such as stylistic bias (Zeng et al., 2024; Raina et al., 2024), length bias (Zheng et al., 2023; Zeng et al., 2024), and positional bias (Wang et al., 2023; Pezeshkpour & Hruschka, 2024). As a result, recent efforts have fine-tuned specialized evaluators Li et al. (2024b); Kim et al. (2024a); Vu et al. (2024), which have been shown to be more robust to common forms of bias (Zhu et al., 2025; Wang et al., 2024a; Park et al., 2024a) while matching the performance of larger prompted models.

---

[*]Work done during an internship at Salesforce AI Research.
[†]Work done at Salesforce AI Research.

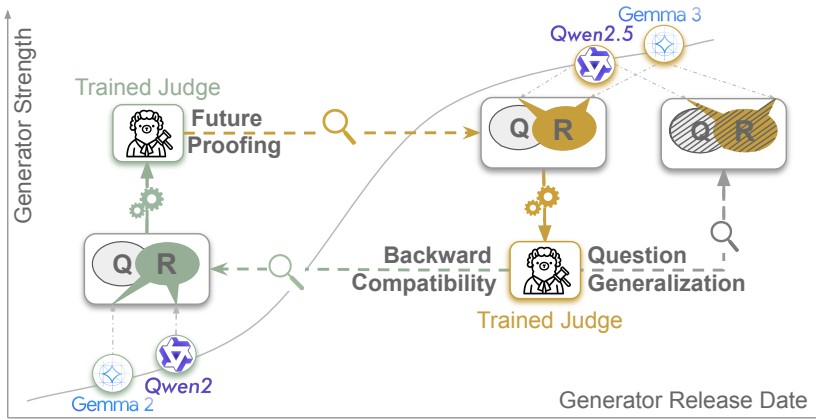

Figure 1: High-level overview of our setup for studying *Future-Proofing*, *Backward-Compatibility*, and *Question Generalization* through the lens of generalization and robustness to input distribution shifts. Q and R represent questions and responses, respectively, with responses generated by the shown generator models (Gemma2, Qwen2, Gemma3, Qwen2.5). *Future-Proofing* evaluates how well judges trained on responses from weaker, older generators (green: Gemma2, Qwen2) assess responses from stronger, newer generators (yellow: Gemma3, Qwen2.5). *Backward-Compatibility* examines the reverse direction. *Question Generalization* measures performance on in-distribution questions and corresponding responses that were both not included (dashed Q and R) in the training.

Although recent advances in judge model fine-tuning have largely focused on developing training methodology Chen et al. (2025a;c), little attention has been devoted to understanding how these models behave as a function of their training inputs. In this work, we investigate this gap by asking three key questions: First, can judge models trained on fixed datasets of input questions, model responses, and ground-truth verdicts accurately evaluate the responses of newer models, i.e., are judges *future-proof*? Second, if we train a judge on up-to-date responses from newer models, can it reliably evaluate responses from older models, i.e., is the trained judge *backward-compatible*? Third, fixing the response generating models, how reliably can judges assess questions that differ from those seen during training, i.e., do they *generalize to new questions*? We examine these questions, as illustrated in Figure 1, through the lens of generalization and robustness, aiming to understand the *shelf life* of trained judges.

In this work, we propose a *dual-distribution* formulation of automatic evaluation. Concretely, we model the judge's input as comprising elements drawn from two distinct distributions: the *question distribution*, which characterizes the input questions to be evaluated, and the *response distribution*, which characterizes the responses to be judged. We study the performance of trained judges when responses are drawn from relatively weak and strong generators, henceforth referred to as weak responses and strong responses. We also examine how well trained judges evaluate questions that are (1) seen during training but paired with new responses, and (2) completely unseen during training. By focusing on weak and strong generators and novel questions, we gain insights into the shelf life of trained judge models through four practical questions:

- **Future-proofing.** Given a judge trained on responses from older ("weak") models, how accurately can it evaluate responses from newer ("strong") models? If the goal is to evaluate strong responses, how much benefit do practitioners gain by training on strong responses rather than weak ones?
- **Backward-compatibility.** Given a judge trained on responses from newer ("strong") models, can it reliably assess responses from older ("weak") models? If the goal is to evaluate weak responses, does training a judge on strong responses provide any benefit?
- **Continual learning.** Compared to judges trained only on weak or strong responses, how well does a continually trained judge adapt to distribution shifts between the two response distributions?
- **Question generalization.** Does judge performance depend on whether a question was seen during training? Even for seen questions, can a judge reliably assess new responses?

Using two verifiable datasets (DeepScaleR and MMLU-Pro), we set up a suite of controlled experiments to analyze the shelf life of judge models, training across three backbone models of varying

sizes and capabilities and three popular judge-training recipes. Our findings reveal that fine-tuned judges struggle to evaluate newer, stronger model responses and therefore require training on an up-to-date response distribution. Once trained on newer, stronger responses, judges exhibit some degree of backward-compatibility. Continual training provides a more balanced adaptation to shifts between older and newer response distributions than training solely on stronger or solely on weaker responses. Finally, we find that fine-tuned judges struggle to generalize to new questions. In all, our findings inform the development and deployment of future generations of fine-tuned judge models.

## 2 BACKGROUND AND RELATED WORK

### 2.1 AN OVERVIEW OF FINE-TUNED JUDGES.

LLM-based judges are automatic evaluators that evaluate LLM outputs given some evaluation criteria. While many judges accommodate different evaluation tasks, such as single rating ("Rate this response on a scale of 1-5") (Hu et al., 2024) or classification ("Is this response appropriate?") (Vu et al., 2024), the dominant evaluation paradigm LLM-based judges are deployed with is *pairwise evaluation*. Here, a judge is given a question and two candidate responses, and tasked with selecting the "better" response according to some criteria. Formally, the judge performs the transformation

$$(Q, R_1, R_2) \quad \longrightarrow \quad (C, \hat{V}), \quad C \text{ optional}, \tag{1}$$

where $Q$ is the question, $R_1, R_2$ are the two candidate responses, $C$ is an optional chain-of-thought explanation, and $\hat{V}$ is the verdict of which response is better. We denote $x = (Q, R_1, R_2) \sim \mathcal{X}$ to be the judge input and $y = (C, \hat{V})$ to be the judge output. Pairwise judges are typically evaluated using accuracy or consistent accuracy, the latter accounting for response-order bias as detailed in Appendix D. Due to its popularity and practicality, pairwise evaluation forms the focus of our study.

Past work in judge fine-tuning uses supervised fine-tuning (SFT) (Li et al., 2024b; Kim et al., 2024b; Zhu et al., 2025), preference optimization methods, like direct preference optimization (DPO) (Wang et al., 2024a; Ye et al., 2024; Saad-Falcon et al., 2024), or more recently, reinforcement learning with verifiable rewards (RLVR) (Chen et al., 2025a;c; Whitehouse et al., 2025; Xu et al., 2025b). Starting from a dataset of $(x, V^\star(x))$ pairs, where $V^\star$ denotes the ground-truth verdict/label, each approach constructs training samples differently: SFT and DPO approaches sample judge outputs from a *teacher model*, then use $V^\star(x)$ to categorize judge outputs as either correct outputs $y^+$ or incorrect outputs $y^-$. Then, the judge is trained on $(x, y^+)$ pairs for SFT and $(x, y^+, y^-)$ triplets for DPO. On the other hand, RL approaches directly make use of the $(x, V^\star(x))$ pairs, omitting the need for teacher model explanations.

### 2.2 RELATED WORK

**Distribution Shifts and Generalization.**   Distribution shift, the mismatch between training and evaluation data, is a long-standing challenge in machine learning (Hendrycks & Dietterich, 2019; Koh et al., 2021). Early computer vision studies demonstrated significant accuracy drops under minor perturbations (Hendrycks & Dietterich, 2019), and WILDS extended this to real-world domain shifts (Koh et al., 2021). In LLMs, the problem is amplified as both data and model capabilities evolve over time (Shi et al., 2025). Recent frameworks explore how models transfer across distributions. *Easy-to-hard generalization* examines whether training on easier tasks transfers to harder ones (Sun et al., 2024), which relates to scalable oversight where only easy tasks can be reliably supervised (Amodei et al., 2016); task-difficulty can be estimated using either model or data-centric measures (Swayamdipta et al., 2020). *Weak-to-strong generalization* investigates improving strong models using supervision derived from weaker ones (Burns et al., 2023). Our setting complements these efforts by focusing on distribution shifts that arise from an *evolving population of generators* and by evaluating how judge models adapt to both weak-to-strong and strong-to-weak shifts.

**Analyzing LLM-as-a-Judge.**   Prior work analyzes systematic judge biases such as positional (Wang et al., 2023; Li et al., 2024b), length (Zeng et al., 2024; Park et al., 2024b), and self-preference (Panickssery et al., 2024; Chen et al., 2025b). Prompt design, instructions, and

scoring format strongly affect reliability (Li et al., 2024a), with pairwise judgments often reducing noise and aligning better with human preferences than pointwise scores (Tripathi et al., 2025; Jeong et al., 2024). Other works have emphasized the importance of carefully selecting reference answers (Krumdick et al., 2025), linking to how generator capabilities influence the judge's inputs (Tan et al., 2025). While most studies consider *static* judges on *fixed* datasets, we instead analyze judges in a dynamic setting where generators change over time, introducing response-distribution shifts that motivate our metrics for *future-proofing*, *backward-compatibility*, and *question generalization*.

## 3 AUTOMATIC EVALUATION AS A DUAL-DISTRIBUTION PROBLEM

We propose a novel formulation of the automatic evaluation problem in terms of two distributions: the question distribution and the response distribution. Concretely, let $\mathcal{Q}$ denote the distribution of questions $Q$, and let $\mathcal{R}$ denote the distribution of responses $R$. For pairwise judges, the input distribution $\mathcal{X}$ therefore takes the form

$$\mathcal{X} = \mathcal{Q} \times \mathcal{R} \times \mathcal{R} \tag{2}$$

The question distribution is defined by characteristics such as semantic content (e.g., domains like medical, legal, finance, scientific, or math) and question difficulty (e.g., difficulty can be defined by pedagogical levels, such as high school vs. olympiad-level math problems). For example, we can consider all questions in GSM8K (Cobbe et al., 2021) to come from the same question distribution, as they are all arguably of similar difficulty and semantic content. The response distribution defines the characteristics of the model responses being evaluated, such as style, capability-specific content, or model-family-specific quirks. We denote the training and test input distributions to be

$$\mathcal{X}^{train} = \mathcal{Q}^{train} \times \mathcal{R}^{train} \times \mathcal{R}^{train} \quad \text{and} \quad \mathcal{X}^{test} = \mathcal{Q}^{test} \times \mathcal{R}^{test} \times \mathcal{R}^{test} \tag{3}$$

respectively. Notably, the two responses come from the same generating model, as described in the data construction details in Section 4.2. Separating the *question distribution* $\mathcal{Q}$ from the *response distribution* $\mathcal{R}$ reflects two real-world sources of shift: (1) the emergence of more capable generators (an evolving $\mathcal{R}$), and (2) the introduction of new questions (an evolving $\mathcal{Q}$). This decomposition allows us to isolate and quantify the impact of each factor on judge performance. In Section 5, we instantiate this framework using the weak response distribution $\mathcal{R}_{weak}$ and the strong response distribution $\mathcal{R}_{strong}$ to simulate a model-development timeline (older, weaker vs. newer, stronger responses and LLMs), along with question splits $Q$ drawn from $\mathcal{Q}$ that are either seen or unseen during training. Informally, weak (strong) responses are drawn from LLMs with relatively low (high) accuracy on questions $Q$; we precisely describe generator strength in Section 4. This instantiation enables us to investigate the four practical questions mentioned in Section 1 regarding the *shelf life* of judges. The specifics of how dual-distribution formalization supports our analysis are detailed in Section 5, with a concise connection provided in Appendix E.

## 4 EXPERIMENTAL SETUP

### 4.1 GAUGING GENERATOR STRENGTH.

We ground our study in two datasets with verifiable solutions: DeepScaleR (Luo et al., 2025) and MMLU-Pro (Wang et al., 2024c). DeepScaleR contains 40K Olympiad-style, reasoning-intensive math problems with gold answers. MMLU-Pro, by contrast, provides verifiable MCQ-style, knowledge-intensive questions spanning 14 diverse domains, including STEM, humanities, social sciences, law, business, psychology, and philosophy, enabling us to study judge shelf-life across a broad range of domains. For generators, we utilize a diverse set of popular instruction-tuned models, listed in Table 1 (in Appendix B). For each generator, we sample 20 responses per question and measure its strength using Pass@1. Pass@1 captures the probability that a uniformly sampled attempt is correct and yields two clearly separated clusters, as shown in Figure 2 and Figure 8 (in Appendix B), where recent or larger models achieve substantially higher scores than smaller or older models. Based on this distinguishable performance difference, we cluster low- and high-performing generators into weak (Gemma-2-9B, Qwen-2-7B, Llama-3.1-8B, Ministral-8B) and strong (Gemma-3-12B, Qwen-2.5-7B, Qwen-2.5-32B, Llama-3.3-70B, Mistral-Small-24B) groups, respectively, and use these clusters to define our response-distribution shifts. Further details on generator selection and strength estimation are provided in Appendix B.

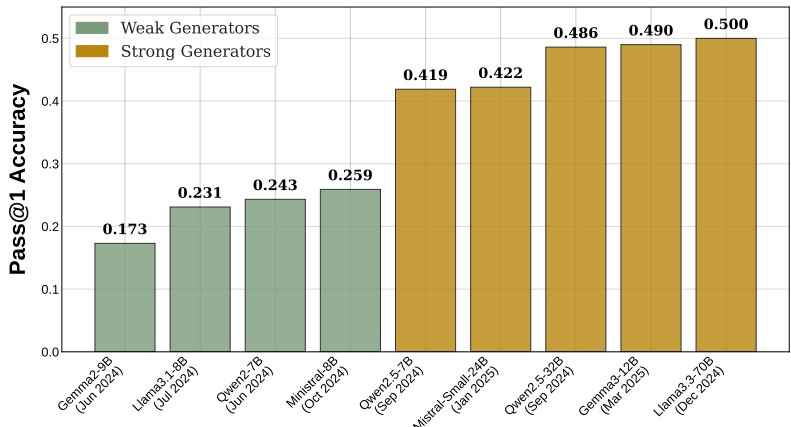

Figure 2: Generator strength on the DeepScaleR dataset, measured using pass@1 with 20 independently sampled responses. Models fall into two well-separated strength clusters: weak (0.17–0.26) and strong (0.42–0.50). No models occupy the 0.26–0.42 gap (a 0.16-wide gap), making the clustering robust to any threshold chosen within this interval. This clustering also aligns with model release dates, with stronger, newer models (yellow) outperforming weaker, older ones (green).

## 4.2 TRAINING SETUP.

**Dataset Construction.** To create the training and evaluation splits, we first construct pairwise input samples for the judge, following prior work (Tan et al., 2025; Wang et al., 2024b). For each question, we sample multiple responses from each generator, and each response is then labeled as "correct" or "incorrect" according to the ground-truth answer $A^\star$. We then form response pairs, where each pair consists of one correct response and one incorrect response, resulting in a pairwise sample with an objectively correct answer. Importantly, responses in a pair are drawn from a single generator only. Based on the generator strengths defined above, we construct datasets of aggregated pairwise samples consisting exclusively of either weak or strong responses, which we refer to as our *weak dataset* and *strong dataset*, respectively.

**Judge Data Distillation & Training Objectives.** We train judges using three commonly adopted recipes: supervised fine-tuning (SFT) (Li et al., 2024b; Kim et al., 2024a; Vu et al., 2024), direct preference optimization (DPO) (Hu et al., 2024; Wang et al., 2024b), and a combined SFT and DPO objective (Wang et al., 2024a; Ye et al., 2024; Saad-Falcon et al., 2024). As these recipes require supervision, specifically, the CoT explanation $C$ (Sec. 2), we adopt the common *teacher model* convention (Li et al., 2024b; Wang et al., 2024a). Based on the ground-truth verdict $V^*$, we categorize responses as correct (positive) samples $y^+$ or incorrect (negative) samples $y^-$. Positive samples are then used for SFT, whereas positive-negative pairs are used for DPO-based recipes.

**Training and Evaluation Splits.** To analyze the four practical questions described in Section 1 using the dual-distribution framework from Section 3, we split the weak and strong datasets into training and test sets. For testing, we construct two distinct splits: an *unseen-questions* split and a *seen-questions* split. The unseen-questions split contains questions not present during training, while the seen-questions split reuses training questions but samples *new* responses, with pairs constructed following the same process as described above. Unless otherwise specified, we use the unseen-questions split for evaluation. We choose three models to train: Llama-3.1-8B, Ministral-8B, and Mistral-24B, covering a range of model sizes and intrinsic strengths.

We provide more details on different aspects of the training setup in Appendix C.

## 5 EXPERIMENTAL RESULTS

In this section, we present our analysis setup and findings on future-proofing, backward-compatibility, and question generalization of judge models. Our analysis builds on the dual-

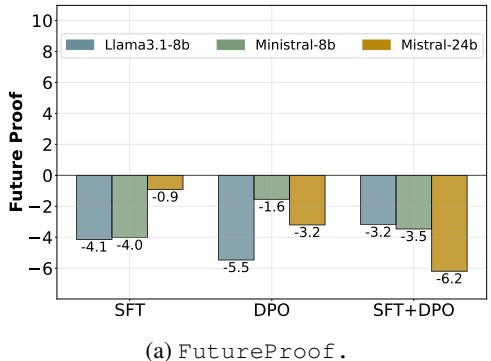 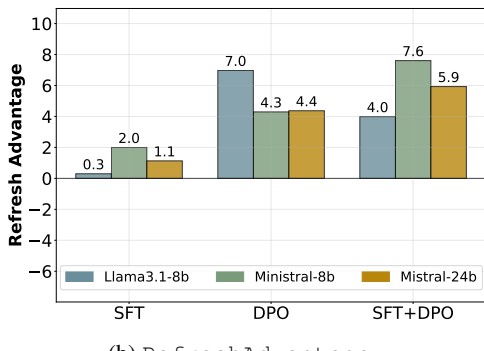

(a) `FutureProof`.      (b) `RefreshAdvantage`.

Figure 3: *Future-Proofing of DeepScaler-Trained Judges.* (a) Future-proofing measured by `FutureProof`; negative values show degraded performance on stronger responses. All models and recipes show performance degradation, indicating poor evaluation of newer, stronger responses. (b) Benefits of re-training on strong responses, measured by `RefreshAdvantage`. Re-training consistently improves performance, with the largest gains under DPO.

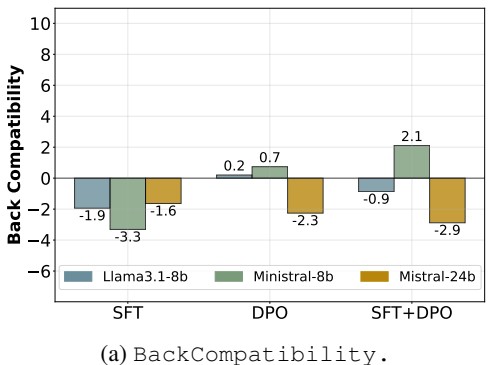 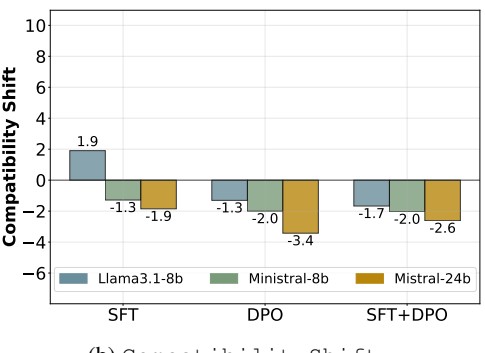

(a) `BackCompatibility`.      (b) `CompatibilityShift`.

Figure 4: *Backward-Compatibility of DeepScaler-Trained Judges.* (a) `BackCompatibility` of judges trained on strong responses when evaluating older responses; positive values indicate improved performance relative to older-judge baselines. Judges trained on newer responses show good `BackCompatibility`, with minimal drops—or even absolute *gains*. (b) Despite strong absolute performance, newer judges still face a distribution shift, reflected by `CompatibilityShift`, with performance drops relative to evaluating strong responses. (c) Compared with future-proofing metrics in Figure 3, backward-compatibility metrics are smaller, indicating that strong-response–trained judges are more backward-compatible than weak-response–trained judges are future-proof.

distribution framework introduced in Section 3, where judge inputs are factorized into a question distribution $\mathcal{Q}$ and a response distribution $\mathcal{R}$. We instantiate the response distribution at two levels of generator strength: $\mathcal{R}_{weak}$ (older, less capable models) and $\mathcal{R}_{strong}$ (newer, more capable models). The question distribution $\mathcal{Q}$ remains fixed but varies in whether a question was seen or unseen during training. In this way, our setup simulates model development timelines. We measure judge performance using consistent accuracy, as defined in Appendix D. Raw consistent accuracy scores are reported in Table 4 of Appendix D, and serve as the foundation for the results below.

**Notation.** For clarity, we denote the consistent accuracy of a judge $J_t$ trained on response distribution $t$ as $\text{Acc}_e(J_t)$, where $t \in \{\text{weak}, \text{strong}\}$. The subscript $e$ indicates the evaluation distribution, with $e \in \{\text{weak}, \text{strong}\}$. Thus, $\text{Acc}_e(J_t)$ ties back to our dual-distribution formalism: it measures the accuracy of a judge trained on distribution $t$ when evaluated on responses from distribution $e$.

## 5.1 How future-proof are judge models?

**Experimental Setup.** To study *future-proofing* in our simulated model development timeline, we design the following setup: weak generators serve as proxies for existing LLMs, and judges are trained on their responses. Strong generators represent newly released LLMs with greater capabilities. By future-proofing, we refer to how well weak-response-trained judges can evaluate responses from newer, stronger LLMs. Specifically, we quantify future-proofing using the following metrics:

`FutureProof` is defined as the difference in the performance of a weak-response-trained judge between the weak and strong evaluation sets:

$$\texttt{FutureProof} = \text{Acc}_{strong}(J_{weak}) - \text{Acc}_{weak}(J_{weak}). \tag{4}$$

This measures the change in performance when the evaluation distribution shifts from $\mathcal{R}_{weak}^{test}$ to $\mathcal{R}_{strong}^{test}$, i.e., a *weak-to-strong* response distribution shift. A positive value indicates relatively better performance on strong responses, while a negative value indicates degradation; thus, higher values correspond to more future-proof judges.

`RefreshAdvantage` is defined as the gain from re-training judges with strong responses:

$$\texttt{RefreshAdvantage} = \text{Acc}_{strong}(J_{strong}) - \text{Acc}_{strong}(J_{weak}). \tag{5}$$

This can be viewed as the *data advantage* from changing the training response distribution from $\mathcal{R}_{weak}^{train}$ to $\mathcal{R}_{strong}^{train}$ when evaluating on $\mathcal{R}_{strong}^{test}$. Higher values indicate greater benefit from re-training judges with the latest and stronger responses.

**`FutureProof` Findings**: For all models and training recipes, we plot the `FutureProof` values on DeepScaleR in Figure 3a. Across all settings, we do not observe any instance where judges generalize to new or stronger responses, with all `FutureProof` values being negative. Interestingly, no discernible trend emerges across training recipes or model families. Generally, we find that SFT leads to greater degradations in smaller models, but a smaller degradation in the large judge. In all, our results show that current judge training approaches do not produce judges capable of reliably generalizing to new, more capable model responses. Beyond lack of generalization, current judge recipes do not exhibit consistent trends across models or scales. These findings align with those on MMLU-Pro, which we discuss in more detail in Appendix F.1. In the absence of recipe-specific or model-specific trends, we recommend evaluating `FutureProof` on a model-by-model basis.

**`RefreshAdvantage` Findings.** Our results on DeepScaleR, presented in Figure 3b, indicate that re-training with up-to-date responses consistently leads to performance gains. In particular, across all training recipes and backbone models, we observe positive `RefreshAdvantage` values. Training recipes also follow a clear trend: retraining with SFT yields minimal but positive gains, whereas DPO yields the largest improvements, providing up to 7.6 absolute percentage points for larger models. The SFT+DPO loss provides additional benefit over DPO alone for a couple of models. We further observe that as judge model size increases, updating training data has a larger impact for DPO-based approaches. For example, with DPO, Mistral-24B exhibits an absolute gain of 7.6 percentage points compared to its 8B counterpart, Mistral-8B, which improves by 4.3 points. These trends are consistent with corresponding findings on MMLU-Pro, discussed further in Appendix F.1. Overall, we conclude that evaluating the most capable models requires training judges on their outputs; relying on stale training data leaves substantial performance gains unrealized.

## 5.2 How backward-compatible are judge models?

**Experimental setup.** Now, we extend our setup for *future-proofing* in Section 5.1 to study *backward-compatibility* in a simulated model development timeline. A judge trained on strong or newer generator responses represents the current judge, which is adept at evaluating new responses, while weak generators represent older LLMs with lower capabilities. By *backward-compatibility*, we refer to how well strong-response-trained judges can evaluate the responses of older, weaker generators. Specifically, we quantify backward-compatibility using the following metrics:

`BackCompatibility` measures the performance gap when evaluating older, weaker responses with the refreshed strong-response-trained judge instead of the weak-response-trained judge:

$$\texttt{BackCompatibility} = \text{Acc}_{weak}(J_{strong}) - \text{Acc}_{weak}(J_{weak}). \tag{6}$$

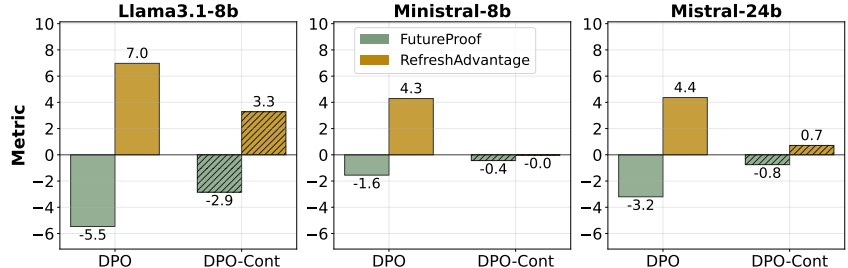

Figure 5: Changes in future-proofing metrics when replacing a weak-response-trained judge (solid) with a continually trained judge (dashed). We observe a decrease in `RefreshAdvantage` and an increase in `FutureProof`, with values approaching zero for a couple of models. This suggests that continual training enables judges to evaluate strong responses more effectively than weak-trained judges, as well as strong-trained judges, and adapts better to the weak-to-strong response shift.

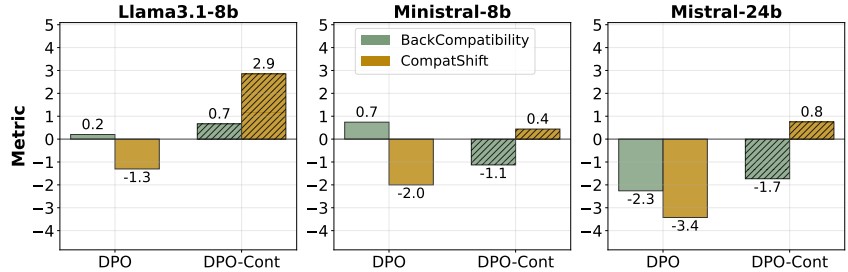

Figure 6: Changes in backward-compatibility metrics when replacing a strong-response-trained judge (solid) with a continually-trained judge (dashed). We see an increase in `BackCompat` for a couple of models, suggesting that continual training can help models better evaluate weak responses than purely strong-trained judges. We also observe an increase in `CompatShift`, showing that continually trained judges adapt better to the strong-to-weak response shift.

This setting is particularly important for established evaluation pipelines: if an old judge is replaced by a new one while the task remains the same, how much does performance differ? We view this as the *data disadvantage* from changing training data from $\mathcal{R}_{weak}^{train}$ to $\mathcal{R}_{strong}^{train}$ when evaluating on $\mathcal{R}_{weak}^{test}$. A positive `BackCompatibility` indicates that the strong-trained judge outperforms the weak-trained judge on weak responses (good backward-compatibility), while a negative value reflects performance degradation (poor backward-compatibility).

`CompatibilityShift` quantifies the weak-to-strong distribution shift when evaluating older, weaker responses with a strong-response-trained, refreshed judge. As noted in the previous section, the reverse shift (strong-to-weak) can strongly affect judge performance. Here, we measure how the out-of-distribution nature of backward-compatibility impacts the newly trained judge:

$$\text{CompatibilityShift} = \text{Acc}_{weak}(J_{strong}) - \text{Acc}_{strong}(J_{strong}). \tag{7}$$

This captures the response-distribution shift opposite to `FutureProof`, i.e., from $\mathcal{R}_{strong}^{test}$ to $\mathcal{R}_{weak}^{test}$ or *strong-to-weak*. It measures how far a strong-trained judge falls below its potential under in-distribution evaluation. A positive value indicates better relative performance on weak responses, while a negative value indicates degradation.

**`BackCompatibility` Findings.** In Figure 4a, we visualize the backward compatibility of judge models trained on strong responses for the DeepScaleR dataset. When evaluating on weak responses, there is little drop in absolute performance between judges trained on strong responses and those trained on in-distribution weak responses. While methods involving SFT consistently cause small performance drops, our results show that DPO training can enable newly trained judges to *outperform* weak-judge models. The drop due to incompatibility is smaller than the advantage gained when moving from weak to strong responses, as noted in the `RefreshAdvantage` findings. This indicates that judges trained on newer responses are indeed backward-compatible: they closely mimic

the performance of weak-trained judges, even in out-of-distribution settings. Likewise on MMLU-Pro, strongly trained judges perform on par with or better than weak-trained judges when evaluating weak responses, as discussed in Appendix F.2. Thus, combined with our findings in Section 5.1, we conclude that re-training with updated responses is universally beneficial: such refreshed judges are not only much better at evaluating new model responses but can also serve as drop-in replacements for their older counterparts with minimal loss in performance.

**`CompatibilityShift` Findings.** As shown above, judges trained on strong responses roughly match the performance of those trained on weak responses when evaluating older responses. Despite strong absolute performance, such newer judges are evaluating under a *strong-to-weak* distribution shift; Figure 4b plots the drop in performance due to this shift on DeepScaleR dataset. Here, we observe that across all judges and recipes, judges still experience degradation due to the out-of-distribution nature of evaluation, with the lone exception being SFT-trained Llama3.1-8B. Surprisingly, here, the largest model, finetuned from Mistral-24B, experiences the largest absolute drops across all training recipes. These findings highlight that, while stronger trained judges can serve as appropriate drop-in replacements for weaker judges, distribution shift causes them to underperform relative to their potential. We see the same pattern on MMLU-Pro, where strong-trained judges also degrade under response-distribution shift, as discussed in Appendix F.2. However, on DeepScaleR, compared to the degradation from the weak-to-strong response-distribution shift (as measured by `FutureProof` in Section 5.1), these degradations are relatively smaller. This suggests that the weak-to-strong evaluation response-distribution shift is a harder setting than strong-to-weak, again highlighting the importance of re-training judges on new model responses.

### 5.3 CAN CONTINUAL TRAINING IMPROVE FUTURE-PROOFING AND BACKWARD-COMPATIBILITY OF JUDGE MODELS?

**Experimental setup.** Sections 5.1 and 5.2 show that training a judge *from scratch* on responses from newer generators is advantageous in evaluations. An alternative is to *continually update* a judge originally trained on older responses by incrementally fine-tuning it on newer, stronger responses. We simulate this continual-learning paradigm by further training $J_{\text{weak}}$ on responses from stronger generators, denoting the resulting model as $J_{\text{weak}\rightarrow\text{strong}}$ (details in Appendix C). All experiments in this section are restricted to training judges on DeepScaler with DPO due to compute constraints.

To assess the effect of continual training, we evaluate $J_{weak\rightarrow strong}$ on both future-proofing and backward-compatibility metrics, comparing its performance against that of the original weakly trained judge and the strongly trained judge, respectively. Specifically, we compare `FutureProof` and `RefreshAdvantage` when replacing $J_{weak}$ with $J_{weak\rightarrow strong}$ in Equations (4)–(5), as shown in Figure 5. We also compare `CompatibilityShift` and `BackCompat` when replacing $J_{strong}$ with $J_{weak\rightarrow strong}$ in Equations (6)–(7), as shown in Figure 6. Together, these comparisons reveal how continual training helps weak judges adapt to future distribution shifts while retaining compatibility with weaker responses, relative to training from scratch.

**Changes in Future-Proofing.** Figure 5 shows that continual training consistently improves future-proofing. `FutureProof` scores increase across all three models, while `RefreshAdvantage` decreases, approaching zero for Ministral-8B and Mistral-24B. The reduction in `RefreshAdvantage` indicates that the benefit of retraining a strong model from scratch, relative to continual training, largely disappears when evaluating stronger responses. At the same time, the higher `FutureProof` scores of $J_{weak\rightarrow strong}$ demonstrate that continual training enables better adaptation to the weak-to-strong distribution shift than simply retaining the weak model.

**Changes in Backward-Compatibility.** Figure 6 shows mixed but informative results on backward-compatibility. `BackCompatibility` scores increase for Mistral-24B and Llama-3.1-8B but decrease for Ministral-8B. Higher `BackCompatibility` indicates that a continually trained judge remains closer to the weakly trained judge when evaluating weak responses, compared to a model trained solely on strong responses. We also observe a notable increase in `CompatibilityShift`, highlighting that continual training improves adaptation to older, weaker responses relative to purely strong-trained models. Together, these results suggest that continual training can better preserve backward-compatibility in several settings while also enhancing adaptability to distribution shifts.

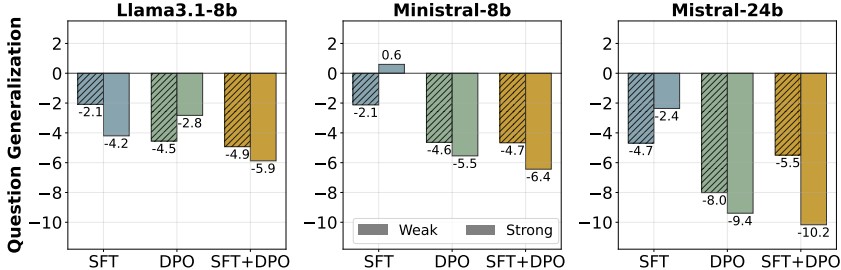

Figure 7: *Question Generalization of DeepScaler-Trained Judges.* Generalization of judges trained on weak vs. strong responses to seen and unseen questions. Judges typically fail to generalize to unseen questions, showing large performance drops relative to evaluating unseen responses on seen questions.

### 5.4 How do judges generalize to unseen questions and responses?

**Experimental setup.** As LLMs advance, both responses and questions evolve (e.g., AIME24 vs. AIME25). We therefore examine how judges perform on previously unseen questions by sampling from $\mathcal{Q}$ in our dual-distribution framework. To quantify the benefits of question exposure during judge training, we define two evaluation splits. In the first, we select a subset of training questions and sample new responses for them, which we call the *seen-questions, unseen-responses* split. In the second, we draw questions from $\mathcal{Q}^{\text{train}}$ that were excluded from training and pair them with new responses, defining the *unseen-questions, unseen-responses* split. Comparing judge performance across these splits reveals the performance gap due to question generalization.

$$\texttt{QuestionGen}_{weak} = \text{Acc}_{weak,unseen}(J_{weak}) - \text{Acc}_{weak,seen}(J_{weak}) \tag{8}$$

$$\texttt{QuestionGen}_{strong} = \text{Acc}_{strong,unseen}(J_{strong}) - \text{Acc}_{strong,seen}(J_{strong}). \tag{9}$$

These metrics capture how well judges generalize across *questions*: responses are drawn from the same generator, with only the question split (seen vs. unseen during training) varied. A positive value of `QuestionGen` indicates better performance on unseen questions, while a negative value indicates failure to generalize to unseen questions.

**`QuestionGen` Findings.** As shown in Figure 7, current judge models do not generalize well to unseen questions, with nearly all judges exhibiting performance drops compared to evaluating on seen questions with unseen responses. Surprisingly, we find that SFT enables the best generalization, with SFT-trained judges showing the smallest absolute drops in most cases. Mistral-24B, however, exhibits the largest drops within each training recipe, indicating poorer generalization compared to smaller models. These trends are consistent with the corresponding findings on MMLU-Pro, discussed in detail in Appendix F.3. Overall, our experiments reveal that exposing judges to the questions they are likely to evaluate can lead to significant performance gains.

## 6 Conclusion

We present a dual-distribution framework for automatic evaluation and analyze four key questions surrounding finetuned LLM-as-judge models, a crucial component of the LLM development cycle. First, we study future-proofing and show that judges trained on older responses struggle to evaluate outputs from newer, stronger LLMs, but re-training on newer responses yields substantial gains. Second, we examine backward-compatibility and find that judges trained on newer responses incur only minor drops, or even improvements, when evaluating older responses. Third, we demonstrate that continual learning provides a more balanced adaptation to both older and newer response distributions compared to training solely on stronger or weaker responses. Finally, we investigate question generalization and find that judges experience large drops in performance on questions unseen during training. Overall, our work highlights critical challenges and actionable strategies for developing robust, future-proof, and backward-compatible judge models.

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

APPENDIX CONTENTS

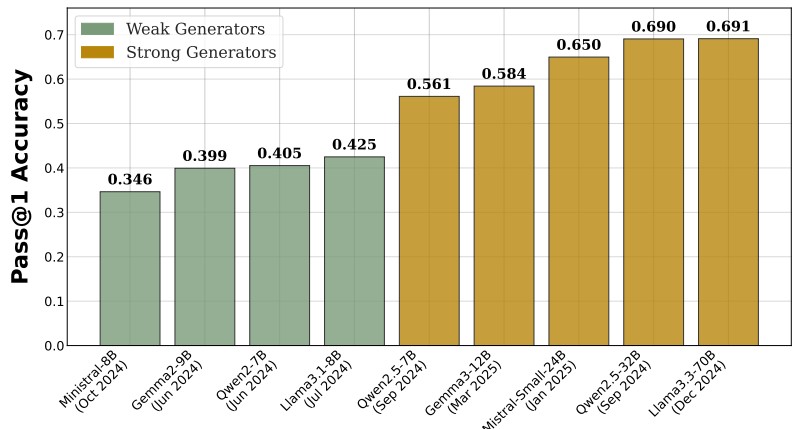

Figure 8: Generator strength on the MMLU-Pro dataset, measured using pass@1 with 20 independently sampled responses. The same two-tier structure emerges: weak (0.34–0.43) vs. strong (0.56–0.69), with no models in the intermediate 0.43–0.56 range (a 0.13-wide gap). This clustering aligns with the weak–strong separation observed in DeepScaleR (Figure 2), suggesting that the pattern reflects underlying model strength rather than threshold choice or dataset-specific artifacts.

## A   LLM USAGE

Other than being used as part of the experiments conducted in this work, LLMs were used solely as a writing assistance tool in preparing this paper submission. Their role was limited to polishing language, improving clarity, and reducing redundancy. The prompt used for this purpose was similar to "Please revise the writing of this, making sure to remove any grammatical mistakes." All research ideas, experimental designs, analyses, and claims presented in the paper are entirely the original work of the authors. No part of the conceptual, methodological, or empirical contributions relies on or originates from LLM outputs.

## B   GENERATORS AND GENERATOR STRENGTHS

| Shorthand | Full Hugging Face Identifier |
|---|---|
| Llama3.3-70B | `meta-llama/Llama-3.3-70B-Instruct` |
| Llama3.1-8B | `meta-llama/Llama-3.1-8B-Instruct` |
| Qwen2-7B | `Qwen/Qwen2-7B-Instruct` |
| Qwen2.5-7B | `Qwen/Qwen2.5-7B-Instruct` |
| Qwen2.5-14B | `Qwen/Qwen2.5-14B-Instruct` |
| Qwen2.5-32B | `Qwen/Qwen2.5-32B-Instruct` |
| Gemma2-9B | `google/gemma-2-9b-it` |
| Gemma3-12B | `google/gemma-3-12b-it` |
| Ministral-8B | `mistralai/Ministral-8B-Instruct-2410` |
| Mistral-Small-24B | `mistralai/Mistral-Small-24B-Instruct-2501` |
| DeepScaleR | `agentica-org/DeepScaleR-Preview-Dataset` |
| MMLU-Pro | `TIGER-Lab/MMLU-Pro` |

Table 1: Mapping from shorthand model and dataset names to their corresponding Hugging Face identifiers.

To curate generator responses, we begin with a set of candidate generators and a collection of questions $Q$, along with verifiable ground-truth answers $A^\star$. For each question, we sample 20 responses from each generator using temperature sampling and compute a *pass@1* score. This score represents the probability of obtaining at least one correct solution when randomly selecting one solution from

| Judge Backbone LLM | Weak Response Dataset | Strong Response Dataset |
|---|---|---|
| Ministral-8B, Mistral-Small-24B | Gemma2-9B, Qwen2-7B, Llama3.1-8B | Qwen2.5-7B, Gemma3-12B, Llama3.3-70B |
| Llama3.1-8B | Gemma2-9B, Qwen2-7B, Ministral-8B | Qwen2.5-7B, Gemma3-12B, Mistral-Small-24B |

Table 2: Overview of training data composition on a per-backbone LLM basis. To mitigate bias from the difficulty of evaluating self-generated responses, we avoid training judge models on their own responses. This produces per-judge training datasets composed of different generator responses.

the 20 samplings, where correctness is determined by matching the generator's responses against $A^\star$.

Concretely, we use two verifiable datasets, as shown in Table 1: DeepScaleR (Luo et al., 2025) and MMLU-Pro (Wang et al., 2024c). DeepScaleR contains 40K challenging Olympiad-level math problems spanning multiple years, each paired with a ground-truth answer. In contrast, MMLU-Pro contains 12K multiple-choice problems, spanning 57 subjects across 14 categories and drawing from diverse sources such as MMLU, STEM websites, TheoremQA, and SciBench. We include MMLU-Pro to demonstrate the broader applicability of our results beyond mathematics. For all experiments, we use popular open-source instruction-tuned models, as listed in Table 1. Gemma-2-9B (Riviere et al., 2024), Gemma-3-12B (Kamath et al., 2025), Llama-3.1-8B, Llama-3.3-70B Dubey et al. (2024), Ministral-8B (Team, a), and Mistral-Small-24B (Team, b), Qwen2-7B (Yang et al., 2024a), Qwen2.5-7B (Yang et al., 2024b), and Qwen2.5-32B (Yang et al., 2024b).

Figure 2, which plots the pass@1 scores of all candidate generators, reveals two clearly separated capability bands on DeepScaleR. Weak models fall in the 0.17–0.26 range, whereas strong models fall in the 0.42–0.50 range, leaving a 0.16-wide empty gap (0.26–0.42) with no model in the intermediate region. Thus, any threshold chosen within this interval produces the same weak-strong grouping. This separation also aligns well with model release dates, as shown in Figure 2. We observe the same two-tier pattern on MMLU-Pro, as shown in Figure 8: weak models score 0.34–0.43, while strong models score 0.56–0.69, again with no models occupying the 0.43–0.56 interval (a 0.13-wide gap). The alignment of weak–strong groups across two very different datasets indicates that the distinction captures genuine differences in underlying model strength, rather than artifacts of a particular dataset or threshold choice.

## C  TRAINING SETUP DETAILS

**Dataset Construction.**   To create the training and evaluation splits, we first construct pairwise input samples for the judge, following prior work (Tan et al., 2025; Wang et al., 2024b). For each question, we sample multiple responses from each generator, and each response is then labeled as "correct" or "incorrect" according to the ground-truth answer $A^\star$. We then form response pairs, where each pair consists of one correct response and one incorrect response, resulting in a pairwise sample with an objectively correct answer. Importantly, responses in a pair are drawn from a single generator only. This choice ensures that the judge learns to distinguish correctness based on reasoning quality rather than relying on stylistic differences between models, which could occur if responses from different generators were mixed in a single pair. For each generator and question, we only keep samples where there is at least one correct and one incorrect response and if this condition is not met, the question is discarded for that generator. In Table 3, we report the percentage of questions retained for each generator after applying this discarding criterion. Further, in Figure 9, we show that weak models discard many hard questions because all 20 samples are incorrect, whereas strong models discard many easy questions because all 20 samples are correct. Mid-tier models retain the most questions because they more frequently produce mixed outcomes, resulting in a clear U-shaped trend in the rank–retention plot. Thus, the retained subset is enriched for borderline questions near each model's decision boundary, naturally inducing a medium-difficulty selection bias. Following this, and based on the generator strengths defined in Appendix B, we construct aggregated pairwise datasets consisting exclusively of either weak or strong responses, which we refer to as the *weak dataset* and *strong dataset*, respectively.

| Generator | DeepScaleR $\text{Ret\%}_{\text{rank}}$ | MMLU-Pro $\text{Ret\%}_{\text{rank}}$ |
|---|---|---|
| Gemma-2-9B | $36.1_1$ | $63.83_2$ |
| Gemma-3-12B | $57.2_8$ | $49.23_6$ |
| Qwen-2-7B | $52.2_3$ | $79.31_3$ |
| Qwen-2.5-7B | $63.7_5$ | $58.75_5$ |
| Qwen-2.5-32B | $62.9_7$ | $43.61_8$ |
| Llama-3.1-8B | $52.2_2$ | $76.38_4$ |
| Llama-3.3-70B | $47.1_9$ | $34.29_9$ |
| Ministral-8B | $62.5_4$ | $62.32_1$ |
| Mistral-Small-24B | $64.6_6$ | $50.73_7$ |

Table 3: Retention percentage ($\text{Ret\%}$) across DeepScaleR and MMLU-Pro for various generators. The subscript rank denotes each model's Pass@1 rank; higher ranks correspond to models with superior performance, as shown in Figure 2 and Figure 8. Retention measures the fraction of questions where a generator produces both a correct and an incorrect sample across 20 generations.

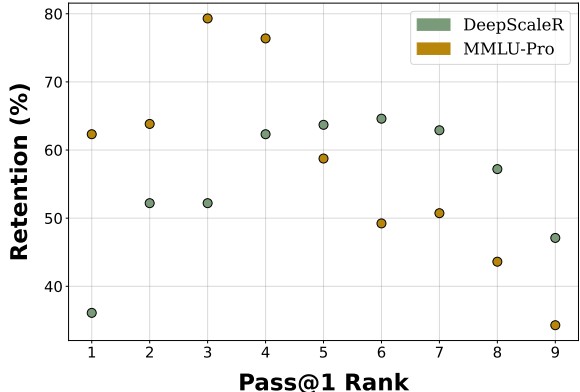

Figure 9: Retention vs. Pass@1 rank, derived from Table 3. Weak generators drop hard questions because all sampled responses are incorrect, whereas strong generators drop easy questions because all sampled responses are correct. Mid-tier generators retain the most questions by producing responses with mixed correctness. As a result, the retained set concentrates on borderline, medium-difficulty items near each generator's decision boundary.

**Judge Data Distillation & Training Objectives.** We train judges using three commonly adopted recipes: supervised fine-tuning (SFT) (Li et al., 2024b; Kim et al., 2024a; Vu et al., 2024), direct preference optimization (DPO) (Hu et al., 2024; Wang et al., 2024b), and a combined SFT and DPO objective (Wang et al., 2024a; Ye et al., 2024; Saad-Falcon et al., 2024). As these recipes require supervision, specifically, the CoT explanation $C$ (Sec. 2), we adopt the common *teacher model* convention (Li et al., 2024b; Wang et al., 2024a). We prompt GPT-4o with $(Q, R_1, R_2)$ inputs, sampling multiple responses $(C, \hat{V})$ per input. Based on the ground-truth verdict $V^*$, we categorize responses as correct (positive) samples $y^+$ or incorrect (negative) samples $y^-$. We only keep inputs for which at least one $y^+$ and $y^-$ exists. This ensures that the inputs are exactly comparable for SFT and DPO. Positive samples are then used for SFT, whereas positive-negative pairs are used for DPO-based recipes.

**Train and Evaluation Splits.** To analyze the four practical questions described in Section 1 using the dual-distribution framework from Section 3, we split the weak and strong datasets into training and test splits. For testing, we construct two distinct splits: an *unseen-questions* split and a *seen-questions* split. The unseen-questions split contains questions not present during training, while the seen-questions split reuses training questions but samples *new* responses, with pairs constructed following the same process as described above. Unless otherwise specified, we use the unseen-questions split for evaluation. Note that the corresponding weak and strong splits use exactly the same set of questions; we remove any question that appears in only one split. This prevents question-

difficulty differences from confounding our findings. Overall, for DeepScaleR each training split contains 70K samples, whereas for MMLU-Pro each contains 10K samples. For both datasets, each evaluation split includes 2.5K response-order-unflipped samples (5K after response-order flips).

**Generator and Judge Backbone Details.** We choose three backbone models to finetune: Llama-3.1-8B, Ministral-8B, and Mistral-24B, covering a range of model sizes and intrinsic strengths. Prior work (Tan et al., 2025) has shown that models often struggle to judge the correctness of pairs of their own sampled responses. Another line of work (Chen et al., 2025b; Panickssery et al., 2024) has shown that models can recognize their own responses and exhibit self-bias. Thus, to disentangle any effects of training a judge on self-generated responses, we exclude the backbone judge model from serving as a generator. Specifically, we create two training sets (each with weak and strong splits), ensuring that the backbone judge model is not included in the list of generators. We summarize these training sets and the associated backbone models in Table 2.

**Hyperparameters.** All experiments with SFT, DPO, SFT+DPO are implemented using the Ax-OLOTL framework Axolotl maintainers and contributors (2023). For SFT, we sweep learning rates in $\{1 \times 10^{-6}, 2.5 \times 10^{-6}, 5 \times 10^{-6}, 1 \times 10^{-5}\}$ with a cosine decay scheduler. Across all evaluation splits, a learning rate of $2.5 \times 10^{-6}$ consistently yields the best performance. For DPO, we adopt standard hyperparameter choices from prior work (Ivison et al., 2023), using a learning rate of $5 \times 10^{-7}$ and a preference strength parameter $\beta = 0.1$. For SFT+DPO, we optimize a joint loss with equal weighting between the SFT and DPO objectives, using the same DPO hyperparameters (learning rate $5 \times 10^{-7}$, $\beta = 0.1$). DeepScaler weak and strong judges are trained for 3 epochs (2,800 gradient steps). In contrast, MMLU-Pro weak and strong judges are trained for 10 epochs (1,500 gradient steps). For continual training experiments (section 5.3), we start from a weak-response DPO-trained judge (trained for 3 epochs) and further train it on strong responses for 1 additional epoch, amounting to roughly 1,000 additional gradient steps. We sweep $\beta \in \{0.1, 1.0\}$ and report results in the main text using $\beta = 1.0$; additional results are included in Table 4 and in Appendix D.

## D  CONSISTENT ACCURACY AND JUDGE'S PERFORMANCE ACROSS SPLITS

**Consistent Accuracy.** Since judge models are prone to positional biases (Wang et al., 2023; Li et al., 2024b; Xu et al., 2025b)—where their preference shifts depending on whether $R_1$ or $R_2$ appears first in the prompt—it is standard practice to evaluate judges using both response orderings (Tan et al., 2025; Xu et al., 2025a;b). Concretely, for input $x = (Q, R_1, R_2)$, let $\bar{x}$ denote the same sample, but with response order flipped in the input prompt, i.e., $\bar{x} = (Q, R_2, R_1)$. Then, evaluation with *consistent accuracy* considers the judge correct only if it correctly identifies the better response under both orderings:

$$\text{Acc} = \frac{1}{|P|} \sum_{x \in P} \mathbb{1}[\hat{V}(x) = V^*(x) \wedge \hat{V}(\bar{x}) = V^*(\bar{x})], \tag{10}$$

where $\mathbb{1}[\cdot]$ is the indicator function, $P$ is the evaluation set consisting of pairs $(x, V^\star(x))$, and the judge's verdicts $\hat{V}(x)$ are compared against the ground-truth verdicts $V^\star(x)$.

**Judge's Performance.** We report all consistent-accuracy scores of our trained judges for both Deep-Scaler and MMLU-Pro across the different evaluation splits in Table 4.

## E  RESEARCH QUESTIONS IN THE DUAL-DISTRIBUTION FORMULATION

As described in Section 3, the dual-distribution formulation separates the *question distribution* $\mathcal{Q}$ from the *response distribution* $\mathcal{R}$, reflecting two real-world sources of shift: (1) more capable generators (an evolving $\mathcal{R}$) and (2) new questions (an evolving $\mathcal{Q}$). This decomposition allows us to isolate and quantify the impact of each factor on judge performance. Building on this, we investigate several practical questions about the *shelf life* of trained judges, focusing on four distinct settings:

**How future-proof are judge models?** For a judge to be future-proof, it must be able to evaluate responses from newer, stronger models. To study this, we examine how a judge trained on responses from the current generation of weak models performs when evaluating responses from strong models. Specifically, we train a judge on $\mathcal{R}_{weak}^{train}$ and evaluate it on both $\mathcal{R}_{weak}^{test}$ and $\mathcal{R}_{strong}^{test}$. This setup

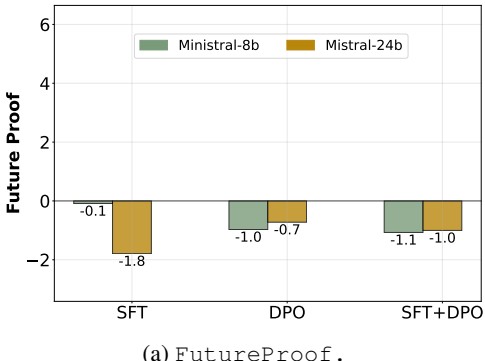

(a) `FutureProof`.

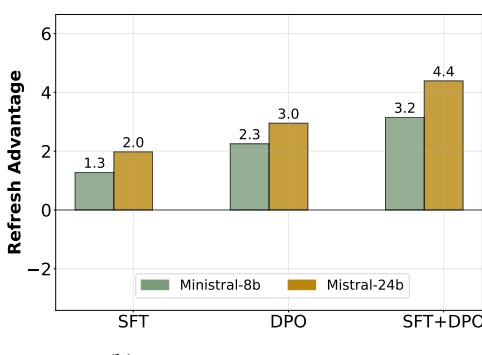

(b) `RefreshAdvantage`.

Figure 10: *Future-proofing of MMLU-Pro–trained judges.* (a) Future-proofing measured by `FutureProof`; negative values indicate degraded performance on stronger responses. All models and training recipes degrade, reflecting poor evaluation of newer, stronger responses. (b) Benefits of re-training on strong responses, measured by `RefreshAdvantage`. Re-training consistently improves performance, with larger gains under DPO-based recipes. These results largely follow the trends observed on DeepScaler (Figure 10), except with smaller absolute magnitudes, indicating that response-distribution shift can depend on the domain.

characterizes how robust judges are to a *distribution shift from weak to strong* responses. Additionally, we quantify the gains from retraining a judge on strong responses by replacing training data from $\mathcal{R}_{weak}^{train}$ with responses from $\mathcal{R}_{strong}^{train}$.

**How backward-compatible are judge models?** Newly trained judges are fine-tuned to evaluate newer, stronger response-generating models. However, does this focus on state-of-the-art generators come at the expense of performance on older, more established generators? To complement our future-proofing experiments, we examine backward-compatibility. Specifically, given a judge trained on responses from $\mathcal{R}_{strong}^{train}$, we ask: how well does it match a judge trained on weaker responses from $\mathcal{R}_{weak}^{train}$ when both are evaluating $\mathcal{R}_{weak}^{test}$ responses? Beyond this comparison, evaluating weaker responses with a judge trained on strong responses also introduces a *distribution shift from strong to weak* responses. We quantify any performance losses that result from this shift.

**Can continual learning improve future-proofing and backward-compatibility of judge models?** Rather than training a new judge from scratch on $\mathcal{R}_{strong}$, we start with a judge trained on $\mathcal{R}_{weak}^{train}$ and continually fine-tune it on $\mathcal{R}_{strong}^{train}$ to obtain a continually trained judge. In parallel to the settings above, we ask whether the continually trained judge narrows the gap on $\mathcal{R}_{strong}^{test}$ relative to one trained only on $\mathcal{R}_{weak}^{train}$, and whether it retains performance on $\mathcal{R}_{weak}^{test}$ relative to a judge trained from scratch on $\mathcal{R}_{strong}^{train}$. This setup tests whether continual training helps a weak judge adapt to the weak to strong response shift while preserving compatibility with older responses.

**How do judges generalize across unseen questions?** As new questions are introduced for evaluating LLMs, judge models must accurately assess responses to these questions. Here, we quantify the benefit of a judge model having seen a question during training. To study this form of *generalization*, we construct two evaluation splits. The first is a *seen-questions, unseen-responses* split, created by selecting questions that appeared in the training set and sampling a new set of responses for these questions from $\mathcal{R}^{train}$. The second is an *unseen-questions, unseen-responses* split, generated by sampling questions from $\mathcal{Q}^{train}$ that were not included in the training data, along with their corresponding responses from $\mathcal{R}^{train}$. Comparing performance across these splits enables us to assess how well judges generalize to previously seen questions versus entirely new ones.

## F    DETAILED FINDINGS FROM MMLU-PRO DATASET

In this section, we present the future-proofing, backward-compatibility, and question-generalization results for the MMLU-Pro dataset and place them in context with the corresponding findings on DeepScaleR. As discussed in Section 5, the overall trends on MMLU-Pro closely match those ob-

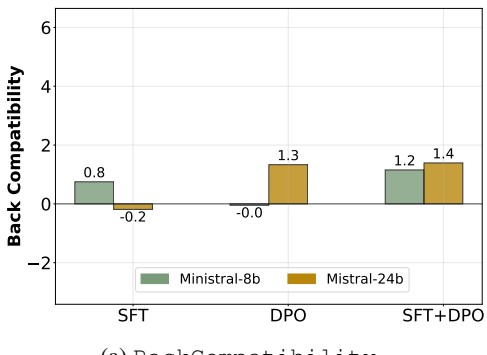 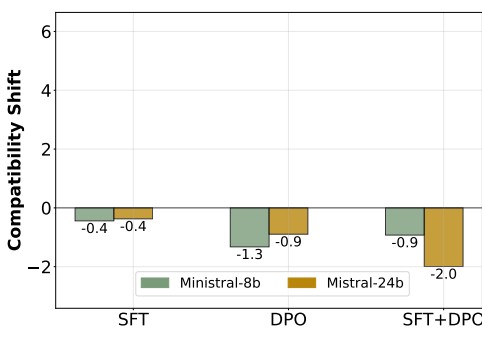

(a) `BackCompatibility`.       (b) `CompatibilityShift`.

Figure 11: *Backward-Compatibility of MMLU-Pro-Trained Judges.* (a) `BackCompatibility` measures how well judges trained on strong responses evaluate older responses; positive values indicate improvements over weak-judge baselines. Strong-trained judges show clear gains, larger than those on DeepScaler Figure 4a, suggesting that strong judges are as good as or better than weak judges when evaluating weak responses. (b) Despite strong absolute performance, newer judges still face distribution shift, reflected in `CompatibilityShift`, which captures performance drops relative to evaluating strong responses. These shifts are similar to those observed on DeepScaler Figure 4b.

Figure 12: *Question Generalization of MMLU-Pro-Trained Judges.* Generalization of judges trained on weak and strong responses to seen and unseen questions. Judges consistently fail to generalize to unseen questions, showing large performance drops relative to their performance on seen questions. These trends align with our findings on DeepScaler dataset in Figure 7.

served on DeepScaler. However, compared to DeepScaleR, which is math-oriented and reasoning-intensive, MMLU-Pro exhibits noticeably smaller degradations across all metrics. Since MMLU-Pro is more knowledge-centered, this suggests that the severity of response-distribution shift is domain dependent. These observations imply that a judge model's shelf-life metrics can vary meaningfully with task domain, even when the training recipe and backbone model are held constant. Below, we describe the results for each metric on MMLU-Pro.

### F.1 How future-proof are judge models?

**`FutureProof` Findings.** Figure 10a reports the `FutureProof` values for all models and training recipes. Consistent with DeepScaleR, we do not observe any case where judges generalize to newer or stronger responses: all `FutureProof` values are negative. However, the magnitudes on MMLU-Pro are noticeably smaller than those on DeepScaleR (see Figure 3a), suggesting that degradation under response-distribution shift is less severe on knowledge-oriented, non-math tasks than on reasoning-intensive math-olympiad problems. This highlights that the extent of future-proofing failure can vary by domain.

**`RefreshAdvantage` Findings.** As shown in Figure 10b, re-training on up-to-date responses consistently improves evaluation performance: all training recipes and backbone models exhibit positive `RefreshAdvantage` values. Mirroring DeepScaleR, DPO-based recipes yield larger gains than SFT alone, and the benefits grow with judge model size. For instance, under SFT+DPO, Mistral-24B gains 4.4 absolute points, compared to 3.2 points for its 8B counterpart, Mistral-8B. Overall, these

results reinforce the DeepScaleR observation that reliably evaluating stronger generators requires judges trained on strong, contemporary responses. However, the gains on MMLU-Pro are slightly weaker compared to DeepScaleR, again indicating that the absolute advantage from refreshing is domain-dependent.

### F.2 HOW BACKWARD-COMPATIBLE ARE JUDGE MODELS?

**BackCompatibility Findings.** In Figure 11a, we visualize the backward-compatibility of judge models trained on strong responses. When evaluated on weak responses, these judges show little improvement over judges trained directly on in-distribution weak responses. In comparison, on DeepScaleR (see Section 5.2), we observed a minimal performance drop, with the `BackCompatibility` metric slightly negative. Taken together, these results indicate that judges trained on newer, stronger responses are indeed backward-compatible: they perform on par with weak-trained judges, even when evaluated out of distribution.

**CompatibilityShift Findings.** Our findings in `BackCompatibility` show that judges trained on strong responses perform comparably to, or better than, weak-trained judges when scoring older responses. However, these stronger judges are evaluated under a *strong-to-weak* distribution mismatch, and Figure 11b illustrates the resulting drop in accuracy. These drops are consistent across models and training recipes. Thus, even though stronger judges can effectively replace weaker ones, distribution shift still limits their realized performance. We observed a similar pattern on the DeepScaleR dataset, as discussed in Section 5.2.

### F.3 HOW DO JUDGES GENERALIZE TO UNSEEN QUESTIONS AND RESPONSES?

**QuestionGen Findings.** From Figure 12, we observe that judges trained on MMLU-Pro do not generalize well to unseen questions, with nearly all judges showing performance drops compared to evaluating on seen questions with unseen responses. The trends are similar to those for DeepScaleR in Figure 7: more performant judges using the DPO recipe and larger backbones such as Mistral-24B exhibit larger drops when evaluated on in-distribution questions not encountered during training.

## G  PROMPTS AND SAMPLING HYPERPARAMETERS

To obtain generator responses, we sample multiple completions from each generator in order to better capture the diversity of its reasoning behaviors. We use five temperature–sampling configurations, where $n$ denotes the number of sampled completions and $t$ the sampling temperature: $(n{=}1, t{=}0.0)$, $(n{=}4, t{=}0.4)$, $(n{=}5, t{=}0.5)$, $(n{=}5, t{=}0.6)$, and $(n{=}5, t{=}0.7)$, with top-$p$ fixed at 1.0. This yields 20 total responses per question for each generator model.

To reduce prompt-format bias and further increase response diversity, we randomly select one of four generator prompt templates (Prompts 1–4) for each sampled completion in the DeepScaleR dataset. For multi-domain experiments using MMLU-Pro, we use the prompt in Prompt 5.

For the judge models, we provide the original question along with two generator responses, each containing both the intermediate reasoning and the final numerical answer. Judge models are decoded greedily using $(n{=}1, t{=}0.0)$, as we found pass@$k$ judge accuracies to be highly correlated with greedy decoding while being more computationally efficient.

We use the prompt in Prompt 6 for judge evaluation.

**Generator Prompt Template 1 — DeepScaleR**

```
Instruction:
Solve the following math problem step by step.  The last line of your
response should be:
Answer: $Answer
where $Answer is the final answer.
Problem:
{{problem}}
Output Format:
Answer: <your answer here>
```

Prompt 1:   Generator prompt template used in DeepScaleR for structured, step-by-step solutions.

**Generator Prompt Template 2 — DeepScaleR**

```
Instruction:
Solve the following math problem efficiently and clearly.
- For simple problems (2 steps or fewer):  give a concise solution with
minimal explanation.
- For complex problems:  use the structured step-by-step format:
  Step 1:  [Concise description]
  [Explanation / calculations]
  Step 2:  [Concise description]
  [Explanation / calculations]
  ...
Important:
Always conclude with:
Therefore, the final answer is: $\boxed {answer}$.
where answer is the final numeric answer.
Problem:
Problem:  {{problem}}
```

Prompt 2:   Generator prompt template used in DeepScaleR that adapts to problem complexity, producing either concise explanations or multi-step structured reasoning.

**Generator Prompt Template 3 — DeepScaleR**

```
Instruction:
Read the problem, reason through it, and provide a final answer.
Problem:
{{problem}}
Output Requirement:
Your response must end with:
The final answer is [answer]
where [answer] is the final computed answer.
```

Prompt 3:   Generator prompt template that prompts models to reason and explicitly report a final answer.

**Generator Prompt Template 4 — DeepScaleR**

```
Problem:
{{problem}}
```

Prompt 4:   A minimal generator prompt template presenting only the raw problem.

---

**Generator Prompt Template — MMLU-Pro**

```
Instruction:
You are given a multiple-choice question from the domain of {{domain}}.
Each answer option corresponds to a lettered choice.
Question:
{{question}}
Options:
{{options}}
Task:
Provide a careful, step-by-step analysis of the question.  Use your
reasoning to evaluate all relevant information and identify the correct
option.  After completing your reasoning, produce your final choice in
the following format:
<answer>X</answer>
where X is the letter of the correct option.
```

Prompt 5:  Prompt template used for reasoning over multiple-choice questions in MMLU-Pro.

---

**Judge Prompt Template**

```
Task:
You are a rigorous evaluator comparing two responses to the same math
question.  Judge which response is better, based solely on logical
soundness and correctness.
You are given:
- A Question
- Response A
- Response B
Evaluation Guidelines:
1.  Correctness is top priority.  Prefer responses with correct
reasoning and correct final answers.
2.  If both have reasoning flaws, choose the one that still reaches the
correct final answer.
3.  Ignore style, length, formatting, verbosity, or fluency.
---
Output Format (JSON):
Your final output must be exactly one of the following:
{"verdict": "A"}
{"verdict": "B"}
---
Question:
{{question}}
Response A:
{{response_a}}
Response B:
{{response_b}}
```

Prompt 6: Judge prompt template used to compare two generator responses based on final-answer correctness and reasoning quality.

| | | DeepScaler | | | | MMLU-Pro | | | |
|---|---|---|---|---|---|---|---|---|---|
| **Train** | **Eval** | **0-Shot** | **SFT** | **DPO** | **SFT+DPO** | **0-Shot** | **SFT** | **DPO** | **SFT+DPO** |
| **Llama3.1-8B** | | | | | | | | | |
| $J_{\text{Wk}}$ | Wk, Sn | 32.44 | 48.14 | 43.95 | 63.40 | – | – | – | – |
| | St, Sn | 28.41 | 44.66 | 36.62 | 60.48 | – | – | – | – |
| | Wk, Un | 30.79 | 46.06 | 39.41 | 58.47 | – | – | – | – |
| | St, Un | 27.76 | 41.91 | 33.94 | 55.29 | – | – | – | – |
| $J_{\text{St}}$ | Wk, Sn | 32.44 | 45.33 | 42.53 | 61.72 | – | – | – | – |
| | St, Sn | 28.41 | 46.41 | 43.74 | 65.15 | – | – | – | – |
| | Wk, Un | 30.79 | 44.12 | 39.61 | 57.60 | – | – | – | – |
| | St, Un | 27.76 | 42.21 | 40.91 | 59.27 | – | – | – | – |
| $J_{\text{Wk}\to\text{St}}^{0.1}$ | Wk, Sn | 32.44 | – | 44.69 | – | – | – | – | – |
| | St, Sn | 28.41 | – | 41.19 | – | – | – | – | – |
| | Wk, Un | 30.79 | – | 40.09 | – | – | – | – | – |
| | St, Un | 27.76 | – | 38.41 | – | – | – | – | – |
| $J_{\text{Wk}\to\text{St}}^{1.0}$ | Wk, Sn | 32.44 | – | 45.43 | – | – | – | – | – |
| | St, Sn | 28.41 | – | 39.13 | – | – | – | – | – |
| | Wk, Un | 30.79 | – | 40.07 | – | – | – | – | – |
| | St, Un | 27.76 | – | 37.22 | – | – | – | – | – |
| **Ministral-8B** | | | | | | | | | |
| $J_{\text{Wk}}$ | Wk, Sn | 33.87 | 48.06 | 61.04 | 61.39 | 26.87 | 34.86 | 47.24 | 47.81 |
| | St, Sn | 28.72 | 41.94 | 55.55 | 56.41 | 27.14 | 34.04 | 46.38 | 47.08 |
| | Wk, Un | 33.81 | 45.93 | 56.41 | 56.72 | 27.05 | 33.72 | 44.54 | 45.03 |
| | St, Un | 29.14 | 41.91 | 54.86 | 53.26 | 25.74 | 33.62 | 43.56 | 43.96 |
| $J_{\text{St}}$ | Wk, Sn | 33.87 | 45.05 | 60.60 | 62.25 | 26.87 | 35.20 | 46.32 | 48.04 |
| | St, Sn | 28.72 | 43.31 | 64.69 | 67.30 | 27.14 | 36.34 | 48.98 | 50.38 |
| | Wk, Un | 33.81 | 42.62 | 57.15 | 58.82 | 27.05 | 34.46 | 44.48 | 46.18 |
| | St, Un | 29.14 | 43.90 | 59.15 | 60.86 | 25.74 | 34.90 | 45.82 | 47.12 |
| $J_{\text{Wk}\to\text{St}}^{0.1}$ | Wk, Sn | 33.87 | – | 62.11 | – | – | – | – | – |
| | St, Sn | 28.72 | – | 60.43 | – | – | – | – | – |
| | Wk, Un | 33.81 | – | 54.67 | – | – | – | – | – |
| | St, Un | 29.14 | – | 53.13 | – | – | – | – | – |
| $J_{\text{Wk}\to\text{St}}^{1.0}$ | Wk, Sn | 33.87 | – | 59.24 | – | – | – | – | – |
| | St, Sn | 28.72 | – | 58.51 | – | – | – | – | – |
| | Wk, Un | 33.81 | – | 55.28 | – | – | – | – | – |
| | St, Un | 29.14 | – | 54.84 | – | – | – | – | – |
| **Mistral-24B** | | | | | | | | | |
| $J_{\text{Wk}}$ | Wk, Sn | 41.00 | 52.18 | 76.57 | 76.90 | 38.51 | 45.14 | 57.14 | 57.53 |
| | St, Sn | 37.69 | 45.34 | 72.16 | 71.94 | 37.17 | 44.81 | 55.25 | 56.42 |
| | Wk, Un | 40.75 | 47.49 | 68.56 | 71.41 | 37.64 | 43.81 | 53.58 | 53.92 |
| | St, Un | 38.03 | 46.57 | 65.36 | 65.21 | 36.25 | 42.02 | 52.86 | 52.92 |
| $J_{\text{St}}$ | Wk, Sn | 41.00 | 47.55 | 73.75 | 75.69 | 38.51 | 44.92 | 56.13 | 57.75 |
| | St, Sn | 37.69 | 50.07 | 79.12 | 81.31 | 37.17 | 46.22 | 59.20 | 61.82 |
| | Wk, Un | 40.75 | 45.85 | 66.30 | 68.52 | 37.64 | 43.62 | 54.92 | 55.31 |
| | St, Un | 38.03 | 47.70 | 69.73 | 71.14 | 36.25 | 43.98 | 55.81 | 57.31 |
| $J_{\text{Wk}\to\text{St}}^{0.1}$ | Wk, Sn | 41.00 | – | 73.70 | – | – | – | – | – |
| | St, Sn | 37.69 | – | 73.80 | – | – | – | – | – |
| | Wk, Un | 40.75 | – | 64.45 | – | – | – | – | – |
| | St, Un | 38.03 | – | 62.37 | – | – | – | – | – |
| $J_{\text{Wk}\to\text{St}}^{1.0}$ | Wk, Sn | 41.00 | – | 78.22 | – | – | – | – | – |
| | St, Sn | 37.69 | – | 75.45 | – | – | – | – | – |
| | Wk, Un | 40.75 | – | 66.83 | – | – | – | – | – |
| | St, Un | 38.03 | – | 66.08 | – | – | – | – | – |

Table 4: *Judge's Consistent Accuracy*. Left block: DeepScaler; right block: MMLU-Pro. *Train* indicates whether the judge is trained on Weak data ($J_{\text{Wk}}$), Strong data ($J_{\text{St}}$), or via continual weak-to-strong training ($J_{\text{Wk}\to\text{St}}^{\beta}$). *Eval* indicates the type of evaluation split defined by the source of responses among Weak (Wk) or Strong (St) and whether questions are Seen (Sn) or Unseen (Un). Within each dataset, columns correspond to the judge-training configurations: Zero-Shot, SFT, DPO, and SFT+DPO. For both datasets, backbone (zero-shot) values are repeated across all blocks to facilitate direct comparison across judge-training strategies.

