# OpenReview forum: "On the Shelf Life of Fine-Tuned LLM-Judges: Future-Proofing, Backward-Compatibility, and Question Generalization"
_ICLR.cc/2026/Conference — ICLR 2026 Poster_

### Official Review · Reviewer_X2CZ · 2025-11-01

**Soundness:** 3
**Presentation:** 3
**Contribution:** 2
**Rating:** 2
**Confidence:** 4

**Summary:**

This paper studies the durability (“shelf life”) of finetuned LLM-as-judge models under realistically inspired distribution shifts. The authors introduce a clear dual-distribution view of automatic evaluation that factorizes a judge’s inputs into a triplet of question and pairwise responses (one correct, other incorrect, but importantly both sampled from the same model). Within this frame, they formalize three practical axes: future-proofing (weak→strong generator shift), backward compatibility (strong→weak), and question generalization (seen→unseen questions). They instantiate the study in math (DeepScaleR, olympiad-style problems) and compare three judge-training recipes (SFT, DPO, SFT+DPO) across three base models (Llama-3.1-8B, Ministral-8B, Mistral-24B). The authors find that: (i) judges trained on “weak” responses do not future-proof—performance drops on “strong” responses; (ii) retraining on “strong” responses yields consistent gains (especially with DPO); (iii) strong-trained judges are fairly backward-compatible on weak responses; (iv) continual training balances both directions better than training from scratch; and (v) question generalization is weak—performance drops notably on unseen questions.

**Strengths:**

### Originality

The paper tackles an important question of *shelf-life* of judge models which I have not seen addressed before in the community. It is a question of increasingly growing importance as the field of LLM-as-a-judge matures. The paper proposes intuitive, measurable quantities—FutureProof, RefreshAdvantage, BackCompatibility, and CompatibilityShift—to quantify judge robustness under evolving generators and questions. These metrics translate deployment concerns into experimentally testable criteria.

### Quality

1. Systematic experimentation: The study spans three judge bases (8B and 24B), three recipes (SFT, DPO, SFT+DPO), and multiple train/test splits (seen vs. unseen questions; weak vs. strong response pools).
2. Clear empirical takeaways: Results are easy to interpret -- future-proofing is hard (negative FutureProof across the board), retraining on strong responses pays off (positive RefreshAdvantage, largest for DPO), strong-trained judges are mostly backward-compatible, continual learning narrows gaps in both directions, and unseen-question performance drops notably.
3. Sound dataset engineering. Responses within a pair come from the same generator to avoid style-based shortcuts; base judge is excluded as generator to reduce self-bias; and response-order flipping is used for evaluation. These are thoughtful choices.

### Clarity

The paper is generally well-structured: motivation→framework→metrics→experiments→implications. Metric definitions are explicit and the content is easy to follow.

### Significance

This work can have practical impact on evaluation pipelines (though currently initiated only in math). The core message—retrain judges with up-to-date strong generator outputs (DPO works best)—is actionable but not surprising. The finding that strong-trained judges can substitute older judges with minimal losses reduces migration risk.

**Weaknesses:**

W1: My main concern is with the fact that authors choose only math as the domain for evaluation. For math, we already have rule-based verifiers to check answers so a judge model is not needed (unless used for step-by-step level check).  Only DeepScaler dataset is used for evaluation. Given that only a single (narrow) domain is studied in the paper, the results may not transfer to more open-ended or subjective tasks (e.g., non-math domains, creative writing). I would consider increasing my score if the authors can add at least 2 other domains (eg: science domains, or mathematical proofs if sticking just to math).

W2: Reliance on a more capable teacher for supervision. SFT/DPO training uses teacher-generated CoT/verdicts; details indicate prompting GPT-4o model for supervisory signals. When studying future compatibility, it is possible that old judges are being used to evaluate frontier models in which case no "stronger" judge exists and it might not be possible to get training data for the judge on responses from strong (newer) models. Thus, the proposed solutions of "retraining on stronger data" or "continual learning" might be infeasible [1]. Do the authors have any approaches for how to address this bottleneck?

W3: The results are mostly expected apriori and not super insightful. For example, the performance drop on “Unseen questions” is not surprising at all. Given that the models were never trained for them, it is expected that the performance of the judges on those questions will be less than the seen ones as they are OOD. Similarly, judge models becoming better with continual learning is also not surprising as that version learns from both strong and weak data compared to other judges which were training only one of the two distributions.

W4: No prompt templates are provided on how the judge model was presented with the responses (was it the whole response or just the final answer?)

W5: Nothing is mentioned about the how the responses were sampled (top_p, temperature values).

## Presentation nits:

1. Some figure y-axes are labeled “Metric” without units; the sign conventions (negative=degradation) require mental mapping. Consider captions with concrete units (percentage-point deltas of consistent accuracy).
2. Table 2 repeats “Base (zero-shot)” blocks per model; consider a single consolidated row to reduce clutter.
3. Fig.1 is hard to follow with the arrows.
4. It is not clarified which models are strong and which models are weak in the main paper. While this is mentioned in the Appendix, this should also be stated in the main paper.

References:

[1] Dorner, F. E., Nastl, V. Y., & Hardt, M. (2024). Limits to scalable evaluation at the frontier: LLM as Judge won't beat twice the data. arXiv preprint arXiv:2410.13341.

**Questions:**

Q1: It is a bit strange that base models were finetuned? Why not use Instruct version of the models? What is the justification for using base models?

---

> ### Author Response · Authors · 2025-11-19
> **We broaden domain coverage with MMLU-Pro and clarify that there is no fundamental bottleneck in creating judge-training dataset.**
>
> Thank you for your thoughtful review and for **willing to raise your score upon experiments on additional dataset**! We appreciate your recognition of the *practical impact of our work, the actionability of our judge-update strategy, and the reduced migration risk demonstrated by strong-trained judges*. We address your specific concerns below:
>
> > C1: My main concern is with the fact that authors choose only math as the domain for evaluation. For math, we already have rule-based verifiers to check answers so a judge model is not needed (unless used for step-by-step level check). Only the DeepScaler dataset is used for evaluation. Given that only a single (narrow) domain is studied in the paper, the results may not transfer to more open-ended or subjective tasks (e.g., non-math domains, creative writing). I would consider increasing my score if the authors can add at least 2 other domains (eg: science domains, or mathematical proofs if sticking just to math).
>
> Thank you for raising this concern.
> We would like to mention that reviewer dFqw **noted that shelf-life decay may be more important and potentially more severe in open-ended tasks like dialogue and summarization**.
> While mathematics provides a clean, verifiable testbed for isolating judge degradation, we agree that assessing shelf life in broader settings is important. However, open-ended tasks are not verifiable and are therefore difficult to use in our experimental setup. To address your concern, **we ran several judge experiments on MMLU-Pro**, which has MCQ-style questions spanning 57 subjects across 14 categories covering STEM, humanities, social sciences, law, business, psychology, and philosophy.
> We **have included these new results in the revised paper**. Notably, we have the following insights:
> - The *same weak/strong generator clustering* observed in DeepScaleR’s math-olympiad setting also appears on MMLU-Pro (Appendix B, lines 963–992).
> - *The overall trends for future-proofing, backward-compatibility, and question-generalization seen in DeepScaleR also appear in MMLU-Pro.* However, compared to DeepScaleR, which is math-oriented and reasoning intensive, MMLU-Pro exhibits noticeably smaller magnitudes across all metrics. Since MMLU-Pro is more knowledge-centered, this suggests that the severity of response-distribution shift is domain dependent. These observations imply that a judge model’s shelf-life metrics can vary meaningfully with task domain, even when the training recipe and backbone model are held constant (Appendix F).
>
> > C2: Reliance on a more capable teacher for supervision. SFT/DPO training uses teacher-generated CoT/verdicts; details indicate prompting GPT-4o model for supervisory signals. When studying future compatibility, it is possible that old judges are being used to evaluate frontier models in which case no "stronger" judge exists and it might not be possible to get training data for the judge on responses from strong (newer) models. Thus, the proposed solutions of "retraining on stronger data" or "continual learning" might be infeasible [1]. Do the authors have any approaches for how to address this bottleneck?
>
> Thank you for raising this question. To perform our analysis, we adopted a controlled setup using verifiable datasets and open-source ‘weak’ and ‘strong’ generators for which inference is inexpensive. However, we agree that this setup omits the scenario where an older judge must learn to evaluate frontier LLM generators without access to even stronger teacher judges.
>
> That said, in this specific scenario **we do not see a fundamental bottleneck in generating the dataset**. We do, however, acknowledge two practical hurdles, detailed below:
> - We clarify that *we do not need an absolutely stronger teacher judge to generate the dataset*, as long as a moderately strong teacher can produce at least one correct judge response across several temperature samplings. This approach is common in recent judge-training work: start with datasets that have verifiable ground-truth answers, generate multiple judge completions from a teacher, bin them into correct or incorrect using the ground truth, and form labeled judge-response pairs. Similarly, RLVR generates such data through on-policy learning. Supervisory signals can therefore be obtained either through rejection sampling from a teacher judge or through on-policy learning. One hurdle is that this requires datasets with verifiable answers, but such data is increasingly available, and RLVR pipelines now routinely scale to millions of verifiable samples.
> - We agree that sampling a frontier-scale generator can be costly, but we do not view this as a deeper limitation. Instead, it reflects the practical cost of producing high-quality, verifiable training data.

---

> ### Author Response · Authors · 2025-11-19
> **We clear the confusion around “base” vs. backbone models, our prompting and sampling setup and explain why our findings are non-obvious.**
>
> > C3: It is a bit strange that base models were finetuned? Why not use the Instruct version of the models? What is the justification for using base models?
>
> We apologize for the confusion. We used "base model" to refer to the judge’s backbone model, but **all models we fine-tune are instruct models, not base models**. We have updated the text to use the "backbone model" for clarity. *Table 1 lists the HuggingFace identifiers for all models and datasets used in our work*.
>
> > C4: No prompt templates are provided on how the judge model was presented with the responses (was it the whole response or just the final answer?). Nothing is mentioned about how the responses were sampled (top_p, temperature values).
>
> To collect generator responses, we sample multiple completions per generator using five temperature-sampling settings (n = number of samples, t = sampling temperature): (n = 1, t = 0.0), (n = 4, t = 0.4), (n = 5, t = 0.5), (n = 5, t = 0.6), and (n = 5, t = 0.7), with top-p = 1.0, yielding 20 responses per question. We also randomly choose one of four prompt templates to increase diversity and better approximate each generator's output distribution.
>
> Judges are provided with the question and two candidate responses, and each response includes both the intermediate reasoning and the final answer. Judge inference uses greedy decoding (t = 0.0, n = 1). Pass@k scores correlate closely with greedy performance, so we report greedy results for efficiency.
>
> **We include all generation and judging prompts and hyperparameters in Appendix F**.
>
> > C5: The results are mostly expected apriori and not super insightful. For example, the performance drop on “Unseen questions” is not surprising at all. Given that the models were never trained for them, it is expected that the performance of the judges on those questions will be less than the seen ones as they are OOD. Similarly, judge models becoming better with continual learning is also not surprising as that version learns from both strong and weak data compared to other judges which were training only one of the two distributions.
>
> We respectfully disagree that the results are simply unsurprising. Although some trends may seem intuitive, **our findings reveal non-obvious behaviors that matter in real judge deployments**, as also noted by other reviewers.
>
> - *Unseen questions*: This setting is not an OOD shift. All problems come from the same math-Olympiad distribution, with a subset held out. The large performance drop therefore reflects poor in-distribution generalization rather than an obvious OOD failure. This was unexpected and has immediate implications for how judges are trusted on new but in-distribution questions.
>
> - *Continual training*: Improvements were not guaranteed. Mixing weak and strong data can easily distort judge behavior or reduce consistency. Our result that one epoch continual update is more robust than a strong-only judge trained for three full epochs is not predictable without running the experiments and provides concrete guidance for maintaining judge quality.
>
> - *Broader patterns*: The magnitude of shelf-life decay, the clear weak-to-strong separation, and the monotonic drop with stronger generators all emerge empirically and were not obvious beforehand.
>
> Together, these findings go beyond intuition and meaningfully inform how judge models should be updated in practice. We will highlight this more clearly in the camera-ready version.

---

### Official Review · Reviewer_YpH3 · 2025-11-01

**Soundness:** 2
**Presentation:** 2
**Contribution:** 4
**Rating:** 6
**Confidence:** 3

**Summary:**

This paper studies the robustness and “shelf life” of finetuned LLM‑as‑judge models. The core contribution is a dual‑distribution formalization. This framing lets the authors isolate and measure three practically important dimensions: future‑proofing (judges trained on older/weaker generator outputs evaluated on newer/stronger outputs), backward compatibility (the reverse), and question generalization (performance on unseen questions). They define clear metrics—FutureProof, RefreshAdvantage, BackCompatibility, and CompatibilityShift—and evaluate across three base models (Llama‑3.1‑8B, Ministral‑8B, Mistral‑24B) and three training recipes (SFT, DPO, SFT+DPO) on a math domain with verifiable labels. This paper provides more key findings and provides a unifying view and actionable guidance for deploying judge models.

**Strengths:**

1. This paper investigates some key questions in the LLM-as-a-Judge domain with interesting findings & takeaways.
2. Extensive and commonly used setups are evaluated and conducted within the math domain.

**Weaknesses:**

1. All experiments are conducted in the maths domain; adding at least one open-ended question would broaden the scope of this paper. This is the major concern of this paper: are we drawing conclusions from a special case?
2. Current experiments only study one round of weak-to-strong. It would be better to scale to multiple rounds.
3. Currently, all experiments only compare effectiveness; adding efficiency consideration would be better.
4. The paper could benefit more from reading if it were reorganized into a more expressive form. As this paper is not a classic paper, it may not necessarily follow the intro-method-experiment setup.

**Questions:**

See cons

---

> ### Author Response · Authors · 2025-11-19
> **We broaden scope with MMLU-Pro, explain the limits of multiple weak-to-strong rounds, highlight the efficiency–effectiveness trade-off, and clarify our content-organization choices.**
>
> Thank you for your encouraging review and for *acknowledging the interesting and actionable nature of our findings and takeaways for LLM judges*. We address your specific concerns below:
>
> > C1: All experiments are conducted in the maths domain; adding at least one open-ended question would broaden the scope of this paper. This is the major concern of this paper: are we drawing conclusions from a special case?
>
> Thank you for raising this concern.
> We would like to mention that reviewer dFqw **noted that shelf-life decay may be more important and potentially more severe in open-ended tasks like dialogue and summarization**.
> While mathematics provides a clean, verifiable testbed for isolating judge degradation, we agree that assessing shelf life in broader settings is important. However, open-ended tasks are not verifiable and are therefore difficult to use in our experimental setup. To address your concern, **we ran several judge experiments on MMLU-Pro**, which has MCQ-style questions spanning 57 subjects across 14 categories covering STEM, humanities, social sciences, law, business, psychology, and philosophy.
> We **have included these new results in the revised paper**. Notably, we have the following insights:
> - The *same weak/strong generator clustering* observed in DeepScaleR’s math-olympiad setting also appears on MMLU-Pro (Appendix B, lines 963–992).
> - *The overall trends for future-proofing, backward-compatibility, and question-generalization seen in DeepScaleR also appear in MMLU-Pro.* However, compared to DeepScaleR, which is math-oriented and reasoning intensive, MMLU-Pro exhibits noticeably smaller magnitudes across all metrics. Since MMLU-Pro is more knowledge-centered, this suggests that the severity of response-distribution shift is domain dependent. These observations imply that a judge model’s shelf-life metrics can vary meaningfully with task domain, even when the training recipe and backbone model are held constant (Appendix F).
>
> > C2: Current experiments only study one round of weak-to-strong. It would be better to scale to multiple rounds.
>
> Certainly, we agree that evaluating multiple weak-to-strong rounds could be informative, such as *observing a continuous trend in how a judge's shelf life varies rather than the step-wise trend we studied*. In our experiments, the chosen weak/strong split reflects the most dominant model-strength shift round, as shown in Figures 7 and 8. And, each additional round requires collecting generator responses, gathering teacher–judge trajectories, and running multiple judge-training and inference pipelines, which is quite expensive. Given these constraints, and because **we already capture the primary strength shift**, we are prioritizing multi-domain evaluations (for example, MMLU-Pro) over finer-grained weak-to-strong transitions. We leave continuous trends in a judge's shelf life for future work.
>
> > C3: Currently, all experiments only compare effectiveness; adding efficiency consideration would be better.
>
> Thank you for the suggestion. Our **primary goal in this work is to study the effectiveness and shelf-life of judges, so we focused on accuracy-based comparisons**. We agree that efficiency is important, and in our setting it is best measured in terms of training epochs, since the main cost comes from retraining a judge on newer generator responses. *We report the training-epoch costs for all judge variants in Appendix E (Hyperparameters)*. Notably, the continually trained judges require only one epoch while achieving better backward-compatibility and future-proofing metrics, whereas judges retrained from scratch on strong-model responses require three epochs yet do not perform better. We will highlight this efficiency–effectiveness trade-off more clearly in the revised version.
>
> > C4: The paper could benefit more from reading if it were reorganized into a more expressive form. As this paper is not a classic paper, it may not necessarily follow the intro-method-experiment setup.
>
> Thank you for the suggestion. While we do not follow a classic introduction-method-experiment structure, we believe the current organization, with metrics and analyses kept within each section, **supports clearer section-by-section reading**. Notably, *reviewer X2CZ also found the content and structure easy to follow*.

---

### Official Review · Reviewer_dFqw · 2025-11-02

**Soundness:** 3
**Presentation:** 3
**Contribution:** 2
**Rating:** 4
**Confidence:** 2

**Summary:**

The paper studies the “shelf life” of finetuned LLM judges under distribution shifts. Concretely, it analyzes (i) future-proofing (weak→strong generators), (ii) backward compatibility (strong→weak), and (iii) question generalization (seen→unseen) in a unified math-domain setup, using pairwise judging data with gold answers and three training recipes (SFT, DPO, SFT+DPO). Main findings: future-proofing is consistently hard; backward compatibility is relatively easy; continual training helps balance the trade-offs; and all judges degrade on unseen questions.

**Strengths:**

1. The paper clearly defines the problem and provides good metrics. This sets a strong foundation for future-proofing the system, maintaining backward compatibility, and generalizing to new types of questions.

2. A key strength is the paper's choice of a verifiable math domain. This decision smartly avoids the common problem of messy or inaccurate data, and their specific evaluation method further strengthens the results by helping to reduce positional bias.

3. The study's approach of separating "weak" and "strong" generators, along with the continual-training analysis, offers practical guidance on how to roll out updates to "judge" models in a real-world setting.

**Weaknesses:**

1. A significant limitation is that the training method (DPO) and the main test (pairwise accuracy) are fundamentally aligned. The model is, in effect, being tested on the same terms for which it was trained. This gives DPO a built-in advantage that doesn't necessarily prove it's a better judge in a general sense. The paper is missing other kinds of tests (like direct scoring) that would be needed to separate this effect and make a fairer comparison.

2. The paper's main conclusions about "shelf life" are drawn entirely from the domain of mathematics. This is a problem because math, with its clear right answers, is not a good stand-in for complex, open-ended tasks like summarization or dialogue. These real-world tasks involve messy data, complex goals, and shifting user preferences, where all factors that would likely worsen shelf-life decay, not lessen it. The paper's claims are too broad for its evidence, which is limited to a single, clean domain and a simulation of time rather than real-world data.

**Questions:**

See weakness

---

> ### Author Response · Authors · 2025-11-19
> **We clarify that DPO is not advantaged by our pairwise evaluation and address domain-scope concerns by extending shelf-life experiments beyond mathematics to the multi-domain MMLU-Pro benchmark.**
>
> Thank you for your thoughtful review and for recognizing *our clear problem formulation and metrics, as well as the practical value of our insights on updating judge-models*. We address your specific concerns below:
>
> > C1: A significant limitation is that the training method (DPO) and the main test (pairwise accuracy) are fundamentally aligned. The model is, in effect, being tested on the same terms for which it was trained. This gives DPO a built-in advantage that doesn't necessarily prove it's a better judge in a general sense. The paper is missing other kinds of tests (like direct scoring) that would be needed to separate this effect and make a fairer comparison.
>
> Thank you for raising this point. Your concern appears to be **a misunderstanding that conflates the pairwise accuracy of judgement task with the pairwise structure of DPO’s training data**. All judges, both DPO and non-DPO, are trained on the same pairwise judgment task, so DPO is not being evaluated on a task it uniquely matches. The true difference lies only in the training data: non-DPO judges use direct labels for response pairs, whereas DPO uses positive and negative samples derived from the teacher’s outputs on the same response pairs. This does not inherently advantage DPO on pairwise accuracy. Moreover, DPO’s objective does not guarantee stronger pairwise accuracy on unseen generators, which is consistent with our empirical finding that DPO and non-DPO judges exhibit similar shelf-life decay. *Pairwise accuracy is therefore not chosen to favour DPO, but rather because it is the standard evaluation framework in alignment, safety, and reward modeling.* We will revise the paper to carefully distinguish between the pairwise task format and DPO’s pairwise supervision (positive and negative) to avoid this confusion.
>
> > C2: The paper's main conclusions about "shelf life" are drawn entirely from the domain of mathematics. This is a problem because math, with its clear right answers, is not a good stand-in for complex, open-ended tasks like summarization or dialogue. These real-world tasks involve messy data, complex goals, and shifting user preferences, where all factors that would likely worsen shelf-life decay, not lessen it. The paper's claims are too broad for its evidence, which is limited to a single, clean domain and a simulation of time rather than real-world data.
>
> Thank you for raising this concern and for **noting that shelf-life decay may be more important and potentially more severe in open-ended tasks like dialogue and summarization**.
> While mathematics provides a clean, verifiable testbed for isolating judge degradation, we agree that assessing shelf life in broader settings is important. However, open-ended tasks are not verifiable and are therefore difficult to use in our experimental setup. To address your concern, **we ran several judge experiments on MMLU-Pro**, which has MCQ-style questions spanning 57 subjects across 14 categories covering STEM, humanities, social sciences, law, business, psychology, and philosophy.
> We **have included these new results in the revised paper**. Notably, we have the following insights:
> - The *same weak/strong generator clustering* observed in DeepScaleR’s math-olympiad setting also appears on MMLU-Pro (Appendix B, lines 963–992).
> - *The overall trends for future-proofing, backward-compatibility, and question-generalization seen in DeepScaleR also appear in MMLU-Pro.* However, compared to DeepScaleR, which is math-oriented and reasoning intensive, MMLU-Pro exhibits noticeably smaller magnitudes across all metrics. Since MMLU-Pro is more knowledge-centered, this suggests that the severity of response-distribution shift is domain dependent. These observations imply that a judge model’s shelf-life metrics can vary meaningfully with task domain, even when the training recipe and backbone model are held constant (Appendix F).

---

### Official Review · Reviewer_iv6U · 2025-11-09

**Soundness:** 3
**Presentation:** 3
**Contribution:** 3
**Rating:** 6
**Confidence:** 4

**Summary:**

This paper studies the “shelf life” of finetuned LLM judges—how well they keep working as generators evolve and as questions change. The authors introduce a dual-distribution view of judge inputs that separates the question distribution Q from the response distribution R and formalize the judge input as X = Q \times R \times R for pairwise judgments. They then operationalize three axes of robustness: future-proofing (weak-trained judges evaluating stronger/newer responses), backward compatibility (strong-trained judges evaluating older/weaker responses), and question generalization (seen vs unseen questions). Experiments are in math, using DeepScaleR questions with verifiable answers, three base judges (Llama-3.1-8B, Ministral-8B, Mistral-24B), and three recipes (SFT, DPO, SFT+DPO), with performance measured by consistent accuracy. Main findings: future-proofing is generally poor (performance drops on newer responses), re-training on strong responses reliably helps (especially with DPO), backward compatibility is mostly fine, continual training narrows the gap between weak-only and strong-only training, and question generalization is limited (drops on unseen questions).

**Strengths:**

Overall, this is a timely and practically important study for anyone deploying LLM judges. The dual-distribution framing is clean and useful, and the metric design aligns well with the questions practitioners actually face. The experimental results are clear and—importantly—actionable (e.g., prefer DPO and keep judges refreshed with up-to-date strong responses). I see this as a strong empirical contribution. With some clarifications on novelty, dataset construction choices, and statistical reporting, I would lean toward acceptance and could increase my score.

**Weaknesses:**

See Questions

**Questions:**

1. You group generators into “weak/strong” by pass@1 bands (~0.25 vs ~0.45). Please spell out thresholds and sensitivity: do conclusions hold if the bands shift or are defined per-dataset/epoch?

2. You only pair a correct and an incorrect response from the same generator to avoid stylistic confounds. This is reasonable; please also quantify how many questions are discarded due to lack of both labels and discuss any induced selection bias toward “borderline” items.

3. Standardize “future-proofing” vs “future proofing” across abstract/body.

---

> ### Author Response · Authors · 2025-11-19
> **We clarify why the weak/strong clustering is robust, quantify the resulting selection bias from our pairwise construction, and standardize terminology throughout the paper.**
>
> Thank you for your positive review and for your **willingness to increase your score upon receiving additional data-construction details and statistics**! We also appreciate your recognition of our work’s *practical relevance, clear dual-distribution framing, and actionable empirical findings*.
> We address your specific concerns regarding the data-construction below:
>
> > C1: You group generators into “weak/strong” by pass@1 bands (~0.25 vs ~0.45). Please spell out thresholds and sensitivity: do conclusions hold if the bands shift or are defined per-dataset/epoch?
>
> Our pass@1 estimates are stable because they are computed from 20 temperature-sampled responses for roughly 40.3K DeepScaleR questions. On DeepScaleR, generators fall into two clearly separated clusters: weak (0.17–0.26) and strong (0.42–0.50). No model lies in the 0.26–0.42 interval, so any threshold within this empty 0.16 gap yields the same grouping. This clustering also aligns with model release dates (Fig. 7).
>
> We repeated the analysis on MMLU-Pro to test generality. The same two clear clusters appear: weak (0.34–0.43) and strong (0.56–0.69), with no models in the 0.43–0.56 gap, showing that our weak–strong clustering is consistent across very different domains (Fig. 8).
>
> Because both datasets exhibit large empty gaps between weak and strong regions, **our grouping is not sensitive to threshold choice and reflects a genuine generator strength divide**. *These clarifications have been added to Appendix B (lines 963–992).*
>
> > C2: You only pair a correct and an incorrect response from the same generator to avoid stylistic confounds. This is reasonable; please also quantify how many questions are discarded due to lack of both labels and discuss any induced selection bias toward “borderline” items.
>
> Pairwise construction requires that a generator produce at least one correct and one incorrect response for a question. Across the 40.3K DeepScaleR questions, generators retain between 36% and 65% of questions depending on strength (Table 3). Weak models (e.g., Gemma-2-9B at 36.1%) discard many hard questions because all 20 samples are incorrect, while strong models (e.g., Llama-3.3-70B at 47.1%) discard many easy questions because all 20 samples are correct. Mid-tier models retain the most questions (63–65%) since they generate mixed outcomes more often. This pattern is visualized with the U-curves in Figure 9. As a result, **the retained set concentrates on borderline questions near each model’s decision boundary**. *We quantify the retention statistics in detail in Appendix C (lines 1006–1012).*
>
> > C3: Standardize “future-proofing” vs “future proofing” across abstract/body.
>
> Thank you for pointing this out. **We have standardized the terminology** and now use the hyphenated form ‘future-proofing’ and ‘backward-compatibility’ consistently throughout the paper.

---

### Author Response · Authors · 2025-12-03
**Authors’ Global Response and Summary of Revisions**

We sincerely thank all reviewers for the time and care put into evaluating our work. We are encouraged that the paper was described as *“practically impactful and relevant,” “actionable,” and having a “clear problem and metric formulation.”* Your suggestions have been extremely helpful in strengthening the manuscript. We have **uploaded a revised draft** with all edits highlighted in green. All line numbers, section references, and figure citations in this rebuttal refer to the updated draft.
Below, we summarize our changes and key discussions:

- **Experiments on an additional dataset** (raised by reviewers dFqw C2, YpH3 C1, X2CZ C1): We add a multi-domain dataset to our experiments and find that it generally follows similar shelf-life trends to the math-olympiad dataset. See details in Appendix B (lines 963–992) and Appendix F.

- **Details about data construction** (raised by reviewers iv6U C1–C2, X2CZ C4):  We report the robustness of the weak/strong generator grouping (Appendix B, lines 963–992), quantify discarded questions for each model during dataset construction (Appendix B, lines 1006–1012), and provide used prompts and sampling temperatures (Appendix F).

- **Clarifications on analysis choices** (raised by reviewers X2CZ C2–C3, dFqw C1):  We reiterate the data-generation process to show that there is no fundamental bottleneck in producing the dataset. We also clarify that we use instruction-tuned LLMs, not base models (Table 1), and address a misunderstanding that conflated pairwise accuracy with pairwise data.

*We have addressed all reviewer concerns*. Please see below for our responses to each comment.

---

### Author Response · Authors · 2025-12-03
**Authors’ Summary of Rebuttal**

Dear Area Chair,

To support the decision-making process, we summarize the current status of our submission below.

We have made substantial revisions in response to reviewer feedback, and a consolidated summary is provided in the public comment **“Authors’ Global Response and Summary of Revisions.”** After the updates, we believe all concerns have been fully resolved.

Key points:

- **Reviewer X2CZ (rating: 2; confidence: 4)** indicated they would **increase their score** if we reported results on an additional dataset. These experiments are now included. The new dataset *corroborates our original conclusions* and displays consistent trends (Appendix B, lines 963–992; Appendix F). We also address their clarification requests and resolve few misunderstandings about our setup.

- **Reviewer iv6U (rating: 6; confidence: 4)** also mentioned that they would **increase their score and lean toward acceptance** upon receiving additional data-construction details and statistics. These details are now provided extensively in Appendices B and C.

- **Reviewer YpH3 (rating: 6; confidence: 3)** similarly asked for results on an additional dataset. These results are now included, and we address all of their remaining clarifications about our setup.

- **Reviewer dFqw (rating: 4; confidence: 2)** also requested another dataset and had few misunderstandings about our methodology. Both concerns are directly resolved in the revision.

After incorporating these revisions, **none of the original concerns materially affect the validity, empirical strength, or contribution of the paper.** The additional experiments support the generalization of our findings across datasets.

Unfortunately, we **were unable to interact with reviewers** in the discussion phase because the additional experiments required substantial compute time, and the discussion period ended before we could upload the results.

Thank you for handling our submission.

Yours sincerely,

Authors of Submission #18969

---

### Meta-Review · Area_Chair_Bzox · 2026-01-06

**Summary:**

### Summary of the paper
This paper studies the shelf life of fine-tuned LLM-judges from three perspectives: future-proofing, backward-compatibility, and question generalization, to evaluate how the judges work well when the model has evolved, and questions are changed over time. The experiments reveal that future-proofing is difficult, backward-compatibility is relatively easy, and fine-tuned judges struggle to question generalization.

### Summary of the reviewers' concerns
Overall, the reviewers' concerns can be summarized as: (1) narrow domain of experiments (i.e., math), (2) necessity of a more complex scenario (multi-round scaling), (3) clarification on DPO settings, (4) weak findings, and (5) presentation (typo, etc)
During the rebuttal phase, the authors addressed the primary concern of the narrow domain issue by presenting their results on MMLU-pro. Given the initial review and rebuttal, the AC's decision is acceptance.

**Reviewer Concerns:**

- The request for an additional dataset from Reviewer X2CZ, Reviewer YpH3, and Reviewer dFqw was successfully addressed.
- The issues with the DPO settings, the complex scenario, and the significance of the main findings are partially resolved.

**Reviewer Scores:**

Given the reviews, rebuttal, and discussion, the AC predicts the revised score if the authors and reviewers had been able to participate in the discussion fully.
- Reviewer iv6U: 6 -> 6.
- Reviewer YpH3: 6 -> 6.
- Reviewer dFqw: 4 -> 4.
- Reviewer X2CZ: 2 -> 4.

---

### Decision · Program_Chairs · 2026-01-26

Accept (Poster)